# Empowered mothers and co-resident grandmothers: Two fundamental roles of women impacting child health outcomes in Punjab, Pakistan

**Rabia Arif**[1]*, **Azam Chaudhry**[2], **Theresa Chaudhry**[1]

1 Department of Economics, Lahore School of Economics, Burki (Lahore), Pakistan, 2 Faculty of Economics, Lahore School of Economics, Burki (Lahore), Pakistan

* rabiaarif106@gmail.com

**Data Availability Statement:** "Complete data are not publicly available, but can be requested from the DHS program upon reasonable request. The IRB-approval procedures for DHS public-use

## Abstract

We show that i) empowered mothers and ii) coresident grandmothers each benefit children's nutritional health measured by height-for-age z-scores (HAZ) and weight-for-age z-scores (WAZ) for age groups 5 years and less. First, using cross-sectional data from the Pakistan Demographic and Health Survey (PDHS) for the year 2017–18, we estimate the impact of empowered mothers on child health outcomes using an instrumental variable approach to correct for endogeneity. Empowerment is measured by two indices: as a sum of the questions that gauge both attitudinal and behavioral dimensions of female agency and also and using multiple correspondence analysis (MCA) for these same questions. Second, we use a fuzzy regression discontinuity design (FRDD) to measure the causal impact of coresident grandmothers on the health outcomes of the children using multiple rounds of the Multiple Indicator Cluster Survey (MICS) from the years 2008, 2011, 2014 and 2018. The difference between the actual ages of the grandmother from the Potential Retirement Eligibility Criteria (PREC) has been used to correct for potential endogeneity. The results show that on average, the weight for age z-scores (WFA) for children under five increases by 0.28 SD with a one-index point increase in mother's empowerment. Similarly, on average, WFA increases by 0.098 SD when grandmothers are present in a household. Finally, we explore heterogeneity in the average effects stated above based upon the gender of the child as well as the wealth and geographic location of the household. The benefits of mothers' empowerment are largely driven by improvements in girls' nutrition as well as children living in rural areas while the presence of grandmothers primarily improves the nutrition of boys, children in rural areas, and children belonging to poor families.

## 1. Introduction

In this paper, we identify two intra-household mechanisms centered around the role of women that can impact the health outcomes of children in a household. We then estimate the

datasets do not allow the respondents, households or sample communities to be identified. To have access to the data, a registered request of a research project must be submitted and approved by the DHS. The instructions for requesting data from the Demographic and Health Survey (DHS) can be found on their website (https://dhsprogram.com/data/Access-Instructions.cfm). Additional data used in this study is publicly available from the Multiple Indicator Cluster Survey (MICS) website (https://mics.unicef.org/surveys). The authors confirm that others would be able to access these data in the same manner as themselves. The authors also confirm that they did not have any special access privileges."

**Funding:** The author(s) received no specific funding for this work.

**Competing interests:** The authors have declared that no competing interests exist.

causal impact of these mechanisms on the health outcomes of Pakistani children measured by the two standard anthropometric measures; height-for-age z-scores (HAZ) and weight-for-age z-scores (WAZ). The first mechanism is the empowerment of mothers in a household, and the second is the presence of grandmothers in a household. While there has been extensive discussions on empowering mothers to enhance the wellbeing of children, evidence supporting the positive causal effect that an empowered mother may have on children is still relatively limited, especially in the case of Pakistan. The second mechanism under analysis is the impact of grandparents on child health; even though a large proportion of children in Pakistan grow up in the same house as their grandparents, the evidence on how this impacts a child's well-being is limited. Our research shows that households with empowered mothers have healthier children, especially girls, while coresident grandmothers improve the nutritional outcomes of children under 5, specifically boys, and children living in poor and rural households.

The nutritional health of children in their younger years can affect their cognitive skills, emotional stability, growth, and economic status in adulthood, which makes childhood malnutrition an important issue. Globally, 144 million children under the age 5 are stunted, and 14.3 million are severely wasted (World Health Organization (WHO), 2020). Although there has been a substantial improvement in health outcomes for children over time, child malnutrition is still prevalent and has significant implications for human capital formation. Additionally, many developing countries like Pakistan have yet to achieve the Millennium Development Goals related to the outcomes of children under the age of 5. According to WHO and United Nations International Children's Emergency Fund (UNICEF), the nutritional health of children in Pakistan is of a significant concern. When compared to its counterparts Bangladesh and India in Fig 1, we see that although there is a declining trend in the prevalence of Pakistani children being underweight and stunted, child stunting in Pakistan is worse than that in India and Bangladesh.

Mother's empowerment is an important channel through which the health of the children can be improved. The literature has found that the empowerment of mothers can be increased through numerous channels such as education, psychological shifts, social shifts, financial improvements, or political changes. Increased empowerment may better equip mothers to take care of their children in which in turn can lead to improvements in child's health

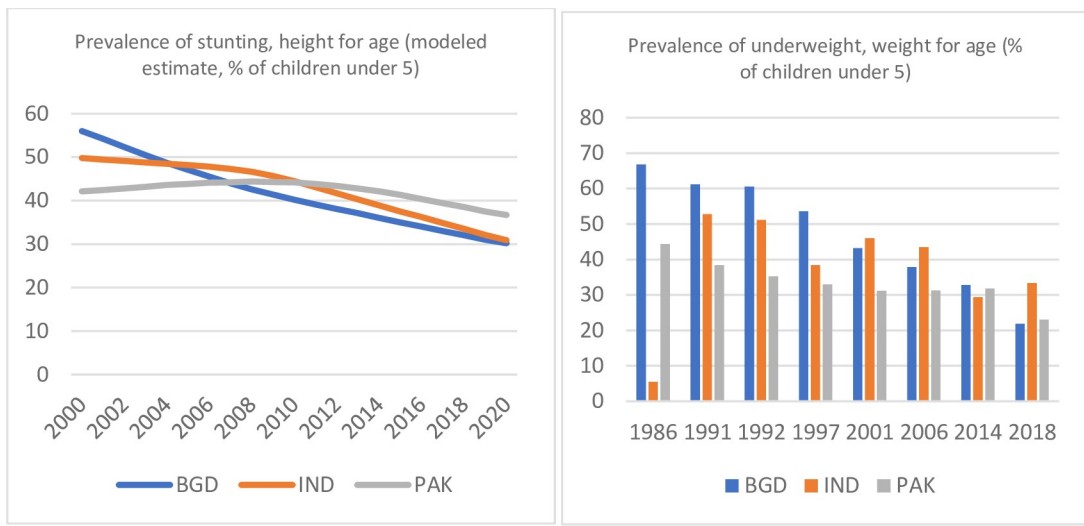

**Fig 1. Prevalence of stunting, height for age and prevalence of underweight, weight for age (modeled estimate, % of children under 5).** Source: The World Bank Database for the years 1991–2020.

outcomes. The positive influence of empowered mothers on the child health outcomes has been extensively documented in the literature [1–3]. However, research has differed in the methods to measure empowerment [1, 4, 5]. In this study, we construct indices to measure the empowerment of mothers in a household that include both intrinsic aspects (i.e., attitudinal dimensions) and the extrinsic aspects (i.e., behavioral dimensions) of mothers [see 6–8]. Furthermore, to address the problem of endogeneity associated with an OLS specification, we instrument for the empowerment of mothers by using the woman's total number of sons. The mother's empowerment index is constructed in two different ways: first, we use the sum of positive responses given, and second, we use a multiple correspondence analysis (MCA) for ten survey questions measuring the behavioral and attitudinal dimensions of daily choices mothers make within the household. Since Pakistan's Demographic Household Survey (PDHS) contains detailed questions regarding these choices for women in addition to the nutritional outcomes of children (along with other important demographic indicators), we use the latest 2017–18 survey, which has data on approximately 5000 children, to estimate our model.

We also look at the impact of grandparents in a household on child nutritional health. The literature has found that while the role of grandparents is less discussed, it may be almost as important for the wellbeing of children as that of parents. This is because of the significant role played by grandparents in the overall caregiving environment because of the financial assistance and emotional support that grandparents can potentially give [9, 10]. As women enter the workforce in greater numbers and as families grow smaller, researchers have recognized the growing importance of grandparents as caregivers for children [11–13]. But much of the previous literature has focused on how factors such as high divorce rates [14, 15], migration [16] and increased childcare expenses [17, 18] have increased the role of grandparents has increased in households. There has been less research on measuring the causal impact of the presence of grandparents in a household on the wellbeing of children. We argue that compared to other (extended) family members, grandparents are more likely to take care of their grandchildren due to both emotional connectedness and the desire to extend the family line, which is sometimes referred to as the "*The Grandmothers Hypothesis*". On the negative side, having more members, such as grandparents, in a joint family structure can affect already constrained resources in a household, which in turn can have negative effects on the health outcomes of children. Thus, the impact of the presence of grandparents is context specific, with some studies finding a net positive impact of the presence of grandparents on the wellbeing of children [19–21], while others have found a negative or insignificant impact [22, 23]. A small number of authors have explored how the impact of grandmothers in a household differs from the impact of grandfathers in a household. Schijner and Smits [15] found that the presence of grandmothers reduced stunting among children, while the presence of grandfathers was more important for girls in poor households for the case of Sub-Saharan Africa. In this research, we argue that the (endogenous) decision of households to include grandparents may bias the measured impact of the presence of grandparents on child health outcomes. Therefore, we use a fuzzy regression discontinuity design exploiting the relationship between the difference between the age of grandparents and the potential retirement eligibility criteria (PREC) with the exogenous presence of grandparents after retirement to measure the causal impact of the physical presence of grandparents on child nutritional health outcomes. Duflo [1] exploits similar variations in data for a cash transfer offered in South Africa extended to pension recipients; her findings found higher weight-for-age and height-for-age for girls (under five years) after the cash-transfer in households that received the pension. These results were driven purely by the pension being offered to grandmothers.

We divide the analysis into (i) measuring the impact of having at least one grandparent in the household, (ii) measuring the impact of having only a grandfather in the household and (iii) measuring the impact of having only the grandmother in the household on the health outcomes of children in a household. We construct a dataset obtained by pooling multiple rounds of UN-funded Multiple Cluster Indicator Surveys (MICS) for 2008, 2011, 2014 and 2018 for this analysis. The constructed dataset contains approximately 200,000 children.

On average, we find that an increase in the additive empowerment index of mothers in a household leads to 0.29 SD increase in the weight-for-age z-scores (WFA z-scores) of children, while an increase in the MCA empowerment index of mothers leads to a 1.168 SD increase in weight-for-age z-scores (WFA z-scores) of children. We find that this impact depends on the location (rural vs. urban) and the gender of the child but is not driven by the wealth of the households. Our results are driven by the positive impact of the empowerment of mothers on girls as well as children living in rural areas. We see a significant increase in WFA z-scores by 0.622 SD for children in rural areas and 0.26 SD significant increase in HFA z-scores for girls.

We also find that on average, the presence of grandparents in a household improves the height-for-age z-scores of children by 0.101 SD and weight-for-age z-scores by 0.0862 SD. However, when we focus on the impact of grandfathers versus grandmothers, we find that the positive results are driven by the presence of grandmothers. Our results show that the increase in weight-for-age z-scores due to the presence of grandmothers primarily accrued to boys, to children in rural areas, and to children in lower wealth families. We see an increase in WFA z-scores of 0.169 SD for boys, an increase in WFA z-scores of 0.160 SD for children in rural areas, and an increase in WFA z-scores of 0.184 SD for children in poor households.

The remainder of the analysis is divided into three sections. The next section discusses our methodology, where we provide a detailed discussion of the data used in the study, the empirical strategy employed, and the descriptive statistics of key variables. We discuss our results in section 3 and conclude in section 4.

## 2. Materials and methods

In this section, we discuss the datasets used in this study, provide details of our empirical strategy, and report the descriptive statistics of key variables in our sample.

### 2.1 Data

We use two datasets to answer our two main research questions. First, to estimate the impact of empowered mothers on the health outcomes of children, we use the Pakistan Demographic Household Survey (PDHS). This is a nationally representative cross-sectional household survey that collects in-depth information on health status, nutritional status, domestic violence against women, education, fertility preferences, and other demographic data. For our analysis, we use data from the 2017–18 PDHS.

Next, we pool multiple rounds of the Multiple Indicator Cluster Survey (MICS) for the years 2008, 2011, 2014, and 2018 to answer the second research question, i.e., whether there is any positive effect of the presence of grandparents on the nutritional health outcomes of grandchildren in a household. We use data from the household surveys conducted in Punjab only, as the data quality from Sindh and Baluchistan provinces was poorer, especially with regards to nutritional indicators; the survey collects detailed information on the location of households, education, nutritional status, employment status, and other demographic information on household members.

Overall, the survey sampling strategies and methodologies used for the PDHS and MICS are similar in many ways, but they differ in terms of the complexity of their sampling methods.

The focus of PDHS is to collect data on reproductive-age women and their health characteristics. It uses a simpler two-stage cluster sampling approach with households used as the primary sampling units (PSU). While the MICS uses a more complex multistage cluster sampling approach and the focus of the data collection is to gather information on the wellbeing of the households. It has been argued that the MICS is more comprehensive and can give more precise estimates.

## 2.2 Empirical strategy

We employ two empirical strategies. First, we use an instrumental variable approach to estimate the impact of mothers' empowerment on child health outcomes using cross-sectional data from the 2017–18 Pakistan Demographic Health Survey (PDHS). Second, we use a fuzzy regression discontinuity design to estimate the impact of the presence of a grandparent in a household on the health outcomes of the children using data from the Multiple Indicator Cluster Survey for the years 2008, 2011, 2014 and 2018.

**2.2.1 Measuring the impact of mother's empowerment on the health outcomes of children.** To measure the impact of the empowerment of mothers on the health outcomes of children, we estimate the following specification:

$$Y_{ghi} = \beta_0 + \beta_1\, MDI_{gh} + \beta_2 Child's\ characteristics_i$$
$$+ \beta_3 Household's\ characteristics_h + \beta_4 Mother's\ characteristics_i + \beta_5 Geographic\ controls_g$$
$$+ \varepsilon_{ghi} \tag{1}$$

where $Y_{ghi}$ is the dependent variable that comprises the nutritional health of child $i$ under age-5 in household $h$ living in district $g$. Two standard anthropometric measures are used to measure the health outcomes of the children: the height-for-age z-score (HAZ) and the weight-for-age z-score (WAZ). The distribution of HAZ and WAZ turns out to be bell-shaped implying that the two dependent variables follow normal distribution (see S1 and S2 Figs). To create a fairer comparison for children of different genders and ages worldwide, the WHO (2010) proposed standardized measures of the height-for-age z-scores and weight-for-age z-scores. The standard formula used to calculate these z-scores is Z = (x-μ)/σ, where x is the original value of the height-for-age and weight-for-age, respectively, μ is the mean of the global reference population set by WHO (2010) and σ is the standard deviation of the original value from the mean of the global reference population (gender and age specific).

The main independent variable, $MDI_{gh}$, is the mother's empowerment index, and we use two different methods to construct the mother's empowerment index. First, we create an additive index of the mother's empowerment, which takes values ranging from 0 to 10 and is constructed by adding up all the responses the woman has given for the ten questions that measure her intrinsic as well as extrinsic level of empowerment. Second, we create a multiple correspondence analysis (MCA) index based upon the same set of questions used for the additive index. The additive index gives equal weight to each respective question whereas the MCA index was generated by assigning ranks to each of the ten qualitative questions according to their relevance and attaches weights to each question to create a weighted sum (see [24] or [25]). In the current literature, principal components (PCA) or factor analysis is most widely used for the construction of such indices. However, PCA is designed to handle quantitative data since it assumes a normal distribution of indicator variables. In contrast, multiple correspondence analysis (MCA) makes fewer assumptions about the underlying distributions of indicator variables and is more suited for qualitative data.

The literature measures the intrinsic level of empowerment by measuring attitudes toward domestic violence. They use women's responses to their level of acceptance about being beaten by her husband in the following situations: 1) if she does not ask permission from her husband when going out, 2) if she gets into an argument with her husband, 3) if she neglects the in-laws or 4) neglects her own children, 5) if she refuses intimacy with her husband, or 6) if she burns the food [see 7, 26]. We similarly assign a value of 1 (0 otherwise) for each time a woman answers that her husband is not justified in beating her in each of the six situations just described.

The literature also looks at the extrinsic level of empowerment, which has been defined as behavioral dimensions of female empowerment, such as the ability of women to exert control over the household's decision-making process, including household purchases, healthcare, and visits to parents [6, 8, 27, 28]. For this, we again assign a 1 (0 otherwise) for each time a woman responds that: 1) she decides on her own about her health care; 2) she decides about her daily purchases; 3) she decides herself to visit her family or relatives; 4) she controls her husband's money.

To construct the empowerment indices, we use the numbers that we obtained from the attitudinal and behavioral measures of empowerment as discussed before. The additive index is simply the sum of the 10 elements so that we obtain a measure of empowerment ranging from zero to 10. The MCA approach is viewed as an extension of principal component analysis where the variables to be analyzed are categorical and not continuous. MCA determines the optimal weights of the empowerment measures and forms a weighted index of empowerment.

Other important variables we control for in the regression are child characteristics that include the child's gender, age, and age squared, and the mother's characteristics, which include the mother's age and age squared. We control for the age squared of the child and the mother in the regression to capture for any non-linearity in the relation between the ages and the health outcomes of children. Additional variables that capture the characteristics of mothers include the age of the first-born child, a dummy equal to 1 for mothers who have ever breastfed, the number of years the mother has been married and a dummy equal to 1 if the mother is pregnant; we also include health inputs which include a dummy equal to 1 if the child is delivered in health care facility, a dummy equal to 1 if the mother has ever used contraceptives and a dummy equal to 1 if the mother had postnatal care at a health facility. Household controls include the age of the household head, age squared of the household head, the gender of the household head, the education of the household head, and the household wealth index. We use location fixed effects by incorporating district controls, provincial dummies, and a dummy equal to 1 if households reside in an urban area.

There are two potential reasons why OLS estimates of the coefficient measuring the impact of the mother's empowerment in Eq (1) may be biased: first, there may be unobserved characteristics of the mother that may also impact the health outcomes of children, and second, there may exist reverse causation between health outcomes of children and the level of empowerment of mothers (i.e., healthier children may lead to greater empowerment of mothers). Therefore, to address this problem, we use the number of sons a woman has as an instrument for mother's empowerment. We argue that the number of sons a woman gives birth to is highly correlated with the empowerment of a woman (especially in the context of South Asia, where sons have been given special importance in a household), but is orthogonal to the health outcomes of children of ages 5 years and less in a household.

The literature provides strong evidence for the link between the number of sons a woman gives birth to and her decision-making power at the household level. While the literature has also found a relationship between female empowerment and a woman having a first-born son, we argue that women in developing countries like Pakistan tend to have more children, which makes the total number of sons a more relevant variable (see [29]) for a more detailed discussion). Alfano [30] argues that women with less control over household incomes secure a

stronger bargaining position by relying more on their male offspring. This point is further reinforced in the literature by the argument that after fathers reach a certain age, mothers gain more power in terms of taking decisions in a household as they become more loyal to the future decision makers in the households, i.e. their sons [6, 31].

For robustness, we also estimated the model using a dummy variable for whether a woman had a first-born son as an instrument for empowerment and found that the first stage for our instrument (the total number of sons) was stronger.

Based on this, we estimate the following first-stage regression:

$$\widetilde{MDI_{hg}} = \gamma_0 + \gamma_1 Number\ of\ Sons\ born\ to\ the\ Mother_h + \gamma_2 \begin{array}{c} Mother's \\ Characteristics \end{array}_m$$

$$+\gamma_2 \begin{array}{c} Child's \\ Characteristics \end{array}_i + \gamma_3 \begin{array}{c} Household \\ controls \end{array}_h + \gamma_4 \begin{array}{c} geographic \\ controls \end{array}_g + u_{hg} \qquad (2)$$

One can argue that having more children in a household might impose stricter resource constraints, which in turn can affect the health outcomes of child $i$ in the age group under 5 years in a household $h$ in district $g$. To address this issue, we add the total number of children as a separate variable in the first-stage regression to control for the impact that increased household size may have on the health outcomes of children directly. We also add other standard variables to control for the impact of constrained household resources.

We estimate the second-stage regression as follows:

$$Y_{ghi} = \sigma_0 + \sigma_1 \widehat{MDI}_{hg} + \sigma_2 Child's\ characteristics_i$$
$$+ \sigma_3 Household's\ characteristics_h + \sigma_4 Mother's\ characteristics_i + \sigma_5 Geographic\ controls_g$$
$$+ \xi_{ghi} \qquad (3)$$

The equation above uses fitted probabilities from Eq (2) to instrument for female empowerment to correct for the endogenous empowerment of mother variable that affects the anthropometric measures of child $i$ in a household $h$ located in district $g$. Again, we show that the distribution of HAZ and WAZ in MICS as well is bell-shaped implying that the two dependent variables follow normal distribution (see S3 and S4 Figs).

**2.2.2 Measuring the impact of the presence of a grandmother on the health outcomes of children.** Next, to measure the impact of the presence of any grandparent, the presence of only grandfathers, and the presence of only grandmothers in a household on the nutritional health outcomes, we use a fuzzy-regression discontinuity design. To estimate this relationship, we estimate the following eq:

$$Y_{ghi} = \beta_0 + \beta_1 G_{gh} + \beta_2 Child's\ characteristics_i$$
$$+ \beta_3 Household's\ characteristics_h + \beta_4 Mother's\ characteristics_i + \beta_5 Geographic\ controls_g$$
$$+ \mu_{ghi} \qquad (4)$$

Where $Y_{ghi}$ is the dependent variable that comprises the nutritional health outcomes for children aged 5 years and younger. We again used two standard anthropometric measures for child health outcomes: the height-for-age z-score (HAZ) and the weight-for-age z-score (WAZ) in a household $h$ with child $i$ located in district $g$.

The variable $G$ is a dummy variable to measure the presence of grandparents. We use three different definitions of $G$ to measure three different potential impacts. First, we define $G$ as

equal to one if at least one grandparent is present in the household. We do this to see the net impact of the presence of one or both grandparents on child health outcomes. Next, we define *G* as equal to one in the case where only the grandfather is present in the household. We do this to see the impact of the presence of only grandfathers on child health outcomes. Finally, we define *G* as equal to one if only the grandmother is present in the household. We do this to see the impact of the presence of only grandmothers on child health outcomes.

Similar to the equation above, OLS estimates of the impact of the presence of grandparents on child health outcomes will be biased. This bias can occur for two reasons. First, families may self-select into choosing whether to live in a nuclear family system or a joint family system. Second, omitted variables, such as values and traditional beliefs, can affect the decision of grandparents to live with their children and their grandchildren and simultaneously affect the health of the grandchildren.

To address this, we use an exogenous threshold, i.e., the retirement eligibility criteria (the methodology is similar to that used in [32], as a potential shifter that can impact the probability of having a grandparent present post retirement age but would not necessarily impact child health outcomes directly. The retirement cutoff age we use for male members of the household was 60 years and for female members of the household was 55 years. The retirement criteria is exogenously fixed for women to be at age of 55 years and for men at 60 years of age over time. We argue that the probability of a grandparent being present in a household increase after this threshold, which allows us to use a regression discontinuity estimation procedure. More specifically, since there is the problem of noncompliance on each side of the cutoff (i.e., grandparents may retire before retirement age or may not retire even if they are above retirement age), we use a fuzzy regression discontinuity design.

The diagrammatic representation of the setup is as follows (a hypothetical scenario). Fig 2 shows a hypothetical scenario that shows if grandparenting had positive impact on health child's health outcomes. Whereas Fig 3 shows a similar hypothetical scenario if grandparenting had a negative impact on the child's health outcomes:

After applying the regression discontinuity framework to our standard regression in Eq (4), we estimate the following set of regressions:

The first-stage regression that we estimate is:

$$\widehat{G_{gh}} = \gamma_0 + \gamma_1 (Age_g - Potential\ Retirement\ Eligibility\ Age)$$

$$+\gamma_2 \frac{Child's}{characteristics_i} + \gamma_3 \frac{Household's}{characteristics_h} + \gamma_4 \frac{Mother's}{characteristics_i} + \gamma_5 \frac{Geographic}{controls_g} + \eta_{ghi} \quad (5)$$

Where $\widehat{G_{gh}}$ is the fitted probabilities of either (i) the presence of at least one grandparent, (ii) the presence of a grandfather only or (iii) the presence of grandmother only in a household as a function of the difference between their respective ages from the retirement criteria and a set of control variables similar to the ones used above, in a household *h* located in district *g*.

The second-stage regression is estimated as follows:

$$Y_{ghi} = \beta_0 + \beta_1 \widehat{G_{gh}} + \beta_2 Child's\ characteristics_i + \beta_3 Household's\ characteristics_h$$

$$+\beta_4 Mother's\ characteristics_i + \beta_5 Geographic\ controls_g + V_{ghi} \quad (6)$$

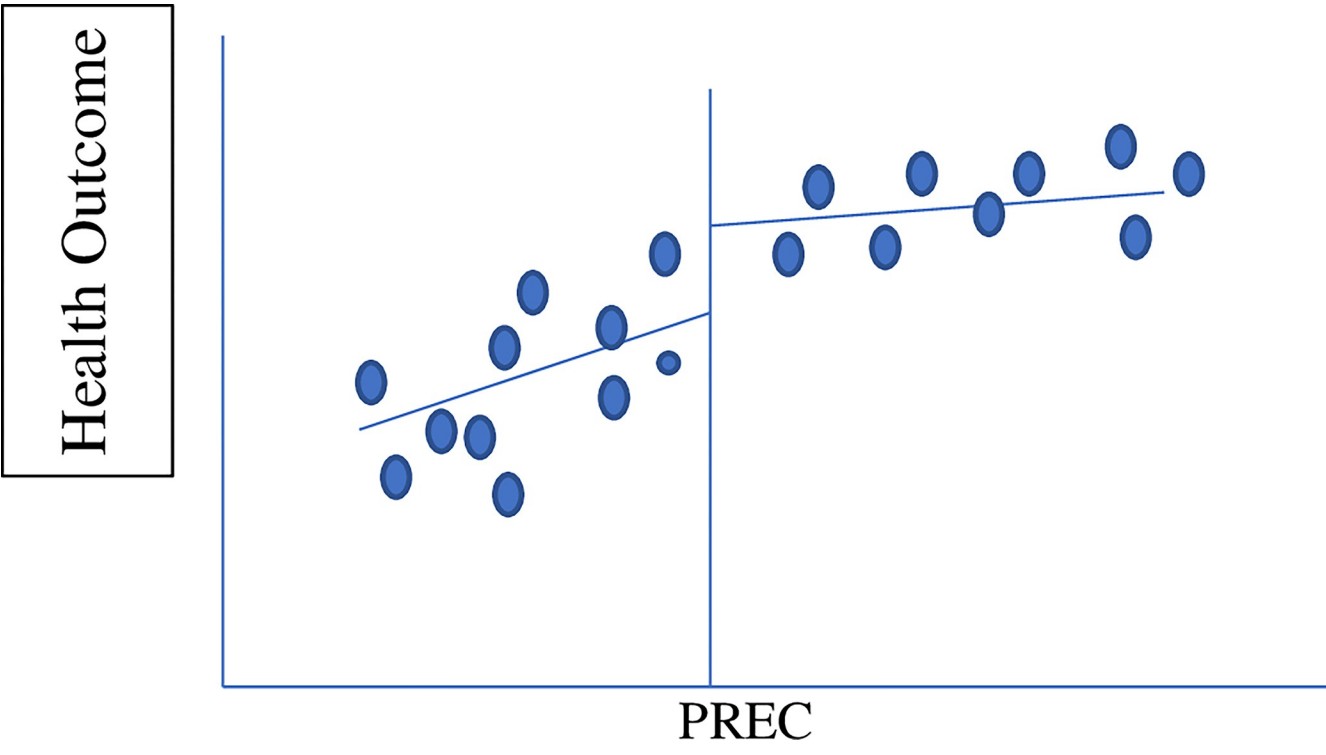

**Fig 2. Positive impact of grandparenting.** Note: RE is the retirement eligibility criteria, which is equal to 60 for male members and 55 for female members according to the global standards.

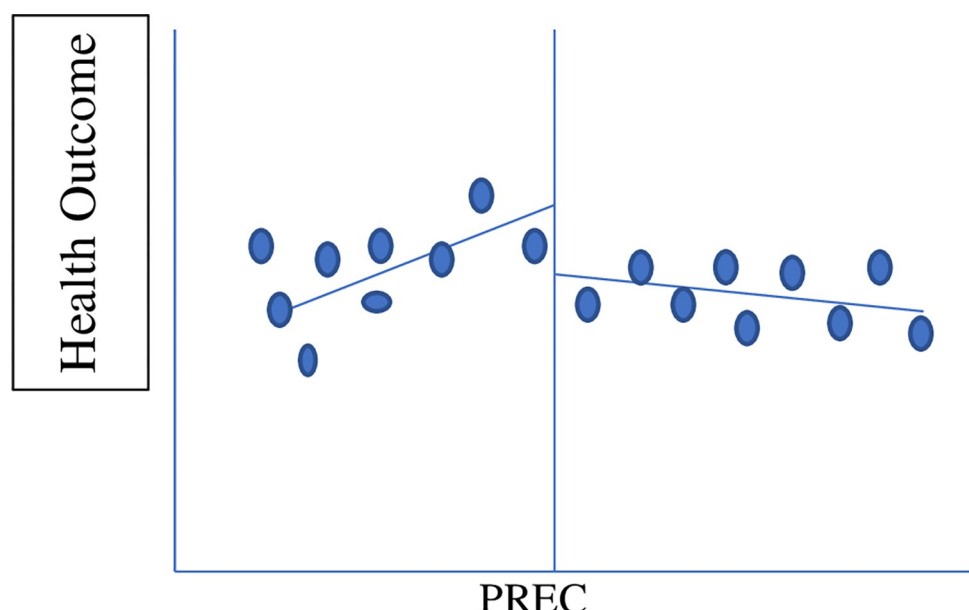

**Fig 3. Negative impact of grandparenting.** Note: RE is the retirement eligibility criteria, which is equal to 60 for male members and 55 for female members according to the global standards.

The equation above is the second-stage regression that uses the fitted values for the probability of having at least one grandparent, or only the grandfather or only the grandmother to correct for endogeneity in a household $h$, with child $i$ in district $g$.

## 2.3 Descriptive statistics

The summary statistics of key variables from the two datasets that have been used in the analyses are discussed in this section. Table 1 reports the descriptive statistics for the cross-sectional dataset of the Pakistan Demographic Household Survey (PDHS) that has been used to estimate the impact of empowered mothers on the health outcomes of the children.

The highlights of the cross-sectional Pakistan Demographic Household Survey (PDHS) for 2017–18 are as follows:

- Out of the total sample of 50,515 individuals, 5,158 are children who lie in the age group of 5 years and less.

- The weight-for-age z-score (WAZ) on average for children (age 5 years and less) in the sample is negative, implying that the weight-for-age of children in households is 1.013 standard deviations lower than the median WAZ of the global reference population (set by WHO standards) of the same age and gender.

- The height-for-age z-score (HAZ) on average for the children (age 5 years and less) in the sample is negative, implying that the weight-for-age of children in households is 1.044 standard deviations lower than the median WAZ of the global reference population (set by WHO standards) of the same age and gender.

**Table 1. Descriptive statistics of key variables from the cross-sectional dataset of the Pakistan Demographic Household Survey (PDHS) year 2017–18.**

| Variable | Observations | Mean | Standard Deviation | Minimum | Maximum |
|---|---|---|---|---|---|
| **Dependent Variables** | | | | | |
| Height-for-age z-scores | 5,158 | -1.013 | 1.010002 | -4.658925 | 5.338705 |
| Weight-for-age z-scores | 4,606 | -1.045 | 1.00848 | -3.829398 | 4.61289 |
| **Child's Characteristics** | | | | | |
| Child's age | 5,158 | 2.435052 | 1.677726 | 0 | 5 |
| Dummy = 1 if child is a boy | 5,158 | 0.5032959 | 0.5000376 | 0 | 1 |
| **Mother's Characteristics** | | | | | |
| Mother's age | 5,158 | 38.47799 | 6.500962 | 16 | 49 |
| Mother's education | 5,158 | 0.8984102 | 1.123097 | 0 | 3 |
| Dummy = 1 if the First born Child is boy | 5,158 | 0.1219465 | 0.3272557 | 0 | 1 |
| If the mother is breast feeding (1 = Yes) | 5,158 | 0.5182241 | 0.4997162 | 0 | 1 |
| Empowerment by Additive Index | 5,158 | 5.536642 | 3.308769 | 0 | 10 |
| Empowerment measured by MCA | 5,158 | 0.060937 | 0.9461473 | -1.497459 | 1.269352 |
| **Household Characteristics** | | | | | |
| Dummy = 1 if the household head is male | 5,158 | 0.8955021 | 0.305935 | 0 | 1 |
| Dummy = 1 if the Household head is Currently working | 5,156 | 0.1183088 | 0.3230047 | 0 | 1 |
| Dummy = 1 if the Households belong to 1st Wealth Quintile | 5,158 | 0.2849942 | 0.4514554 | 0 | 1 |
| Dummy = 1 if the Households belong to 2nd Wealth Quintile | 5,158 | 0.2161691 | 0.4116708 | 0 | 1 |
| Dummy = 1 if the Households belong to 3rd Wealth Quintile | 5,158 | 0.1929042 | 0.3946167 | 0 | 1 |
| Dummy = 1 if the Households belong to 4th Wealth Quintile | 5,158 | 0.1613028 | 0.3678457 | 0 | 1 |
| Dummy = 1 if the Households belong to the 5[th] Wealth Quintile | 5,158 | 0.1446297 | 0.3517612 | 0 | 1 |

**Source:** Author's Own Calculations.

- On average, 50.3% of children in the sample aged 5 years and younger were boys, and the average age of the children from the sample in this age group was approximately 2 years.

- On average, 89.84% of the mothers in the sample had been enrolled in school and are considered educated.

- On average, 12.19% of mothers had a son as their first-born child.

- On average, 51.8% of the mothers breastfed their infants.

- On average, the total number of people in a household was 9 members.

- A total of 89.6% of the households in the sample were male headed.

- A total of 11.83% of the households in the sample had household heads that were employed in the market.

Next, in Table 2 we report the summary statistics of the cross-pooled Multiple Indicator Cluster Survey for data years 2008, 2011, 2014 and 2018 that has been used to estimate the impact of grandparenting on the child's health outcomes.

The summary statistics of key variables from the pooled dataset that was obtained from the Multiple Indicator Cluster Survey (MICS) for 2008, 2011, 2014 and 2018 found the following:

- The average weight-for-age z-score (WAZ) on average for children ages 5 years and less was negative, implying that the weight-for-age of children in households is 1.1485 standard deviations lower than the median WAZ of the global reference population (set by WHO standards) of the same age and gender.

- The average height-for-age z-score (WAZ) for children ages 5 years and less was negative, implying that the weight-for-age of children in households is 1.1402 standard deviations lower than the median WAZ of the global reference population (set by WHO standards) of the same age and gender.

- Of the households in the sample, 16.7% had both grandparents present, 13.4% had only grandfathers present, and 3.3% of the households had only grandmothers present.

- A total of 48.7% of the children in the sample aged 5 years and less were girls, and the average age of these younger children was approximately two and a half years.

- A total of 5.5% of households in the sample were headed by females, the average household size was 8 members, and the average years of education of the household head was 2 years.

- The average age of mothers in the sample was 30 years, the average years of education of mothers in the sample was 2 years, on average, the mothers had been married for 10 years, and the average age of the first-born child in the sample household was 7 years.

- A total of 50.5% of the households in the sample had at least 3 children, and 23.5% had 2 children.

Finally, in Table 3, we explore the differences in the means of weight-for-age and height-for-age between (i) households with at least one grandparent and households without any grandparent, (ii) households with a grandfather and households without a grandfather and (iii) households with a grandmother and households without a grandmother, as well as a t-test of the significance of these differences.

Table 3 shows that the height-for-age z-scores and weight-for-age z-scores of younger children were significantly higher in households with grandparents, grandfathers (alone) or grandmothers (alone) than in households without any grandparent.

**Table 2. Descriptive statistics of key variables from the cross-pooled Multiple Indicator Cluster Survey (MICS) from 2008–2018.**

| Variable | Observations | Mean | Standard Deviation | Minimum | Maximum |
|---|---|---|---|---|---|
| **Dependent Variables** | | | | | |
| **Height-for-age z-scores** | 184124 | -1.485 | 1.608658 | -5.99 | 5.99 |
| **Weight-for-age z-scores** | 187918 | -1.40 | 1.269045 | -5.99 | 5.98 |
| **Independent Variables** | | | | | |
| **Dummy = 1 if only grandfather is present** | 184,124 | 0.1374997 | 0.3443751 | 0 | 1 |
| **Dummy = 1 if only grandmother is present** | 184,124 | 0.0337924 | 0.180695 | 0 | 1 |
| **Dummy = 1 if both grandparents are present** | 184,124 | 0.1712922 | 0.3767651 | 0 | 1 |
| **Child's Characteristics** | | | | | |
| **Age of the child** | 184,124 | 1.987074 | 1.417465 | 0 | 4 |
| **Dummy = 1 if the child is a girl** | 184,124 | 0.4883937 | 0.4998666 | 0 | 1 |
| **Household Characteristics** | | | | | |
| **Families with female household head** | 184,121 | 0.0567453 | 0.2313559 | 0 | 1 |
| **Household Head's Education** | 183,893 | 2.187919 | 1.479037 | 0 | 5 |
| **Mother's Characteristics** | | | | | |
| **Mother's Education Level** | 184,068 | 2.148087 | 1.409148 | 1 | 5 |
| **Age of the mother** | 163,542 | 30.39269 | 5.977785 | 15 | 50 |
| **Age of the first-born child** | 171,271 | 7.330435 | 5.235559 | 0 | 39 |
| **Number of Years Married** | 120,760 | 9.775671 | 5.590754 | 0 | 43 |
| **Dummy = 1 if the Mother Ever breast fed** | 150,243 | 0.7891549 | 0.40791 | 0 | 1 |
| **Dummy = 1 if the mother received postnatal care** | 150,457 | 0.2172714 | 0.4123902 | 0 | 1 |
| **Dummy = 1 if the mother ever used contraceptives** | 182,610 | 0.1594747 | 0.3676328 | 0 | 1 |
| **Dummy = 1 if the mother is Pregnant** | 182,819 | 0.1216504 | 0.3268825 | 0 | 1 |
| **Mother's age at the first-born child** | 163,523 | 27.88362 | 5.818547 | 10 | 50 |
| **Families with 2 or more children** | 184,124 | 0.2329408 | 0.422706 | 0 | 1 |
| **Families with 3 or more children** | 184,124 | 0.5128718 | 0.4998356 | 0 | 1 |
| **Families with 2nd child being girl** | 184,124 | 0.1141731 | 0.3180222 | 0 | 1 |
| **Families with 3rd child being a girl** | 184,124 | 0.2505051 | 0.4333051 | 0 | 1 |
| **Wealth Quantile** | | | | | |
| **Family's belonging to lowest wealth quintile** | 184,124 | 0.210456 | 0.4076336 | 0 | 1 |
| **Families belonging to second wealth quintile** | 184,124 | 0.2017879 | 0.4013358 | 0 | 1 |
| **Families belonging to third wealth quintile** | 184,124 | 0.2076535 | 0.4056285 | 0 | 1 |
| **families belonging to fourth wealth quintile** | 184,124 | 0.2083053 | 0.4060974 | 0 | 1 |
| **Families belonging to highest wealth quantile** | 184,124 | 0.1717973 | 0.3772052 | 0 | 1 |

**Source:** Author's Own Calculations.

## 3. Results

We begin by presenting the results for the specification that tests the relationship between the empowerment of mothers and the health outcomes of younger children as measured by two anthropometric measures: height-for-age and weight-for-age. It is worth noting that height-for-age is considered to measure health outcomes over a longer period, and weight-for-age is considered to be a measure of short-term health. After this, we present the results for the specifications that measure the relationship between the presence of grandparents more specifically grandmothers and nutritional health outcomes of younger children.

**Table 3. Inferential and descriptive statistics of child's health outcomes from cross-pooled Multiple Indicator Cluster Survey (MICS) year 2008–2018, by the presence of at least one grandparent, only grandfather and only grandmother.**

| Dependent Variables | HHs with Grandparents | | HHs without Grandparents | | Difference Between the Mean Values |
|---|---|---|---|---|---|
| | Observations | Mean (1) | Observations | Mean (2) | Mean (1)—Mean (2) |
| **Height-for-age (z-scores)** | 31539 | -1.3647 | 152585 | -1.5104 | 0.1457*** |
| **Weight-for-age (z-scores)** | 32197 | -1.3025 | 155721 | -1.4227 | 0.1202*** |
| | **HHs with Grandfathers** | | **HHs without Grandfathers** | | |
| **Height-for-age (z-scores)** | 25317 | -1.3589 | 158807 | -1.5056 | 0.1468*** |
| **Weight-for-age (z-scores)** | 25865 | -1.2991 | 162053 | -1.4185 | 0.1194*** |
| | **HHs with Grandmothers** | | **HHs without Grandmothers** | | |
| **Height-for-age (z-scores)** | 6222 | -1.3886 | 177902 | -1.4888 | 0.1002*** |
| **Weight-for-age (z-scores)** | 6332 | -1.3164 | 181586 | -1.4051 | 0.0887*** |

Note

*** $p < 0.01$

** $p < 0.05$

* $p < 0.1$

## 3.1 Measuring the impact of mothers' empowerment on health outcomes

In this section, we report the results for measuring the impact of mothers' empowerment on the two anthropometric measures: height-for-age and weight-for-age for children of the age group 5 years and less. As discussed above, we constructed two different indices to measure the empowerment of mothers in a household based upon ten questions reported in DHS for survey year 2018. We start by presenting the first-stage results and then move to the second-stage results. We then explore the heterogeneity in the impact of empowered mothers by creating sub-groups in our sample and present results from regressions that test to see if there are differences in this relationship based on geographic location of households, based on the genders of the children, and across the wealth distribution. In the case of PDHS, the wealth index is created using Principal Component Analysis (PCA) that use various indicators like household characteristics, durable goods and assets to determine the pattern of wealth amongst the households. They later divide the households amongst wealth quintiles. The lowest wealth quintile represents the 20% of the population that is part of the most constrained households (i.e., poorest). We use the bottom two quintiles to identify the population that is severely constrained and most severely constrained in resources and later argue that these households may have been affected differently by the empowerment of mothers.

**3.1.1 First-stage results.** Table 4 gives the first-stage results. We report the results for both indices of empowerment. In both cases, we instrument the women's empowerment indices with the total number of sons in a household. Specifications (1) and (3) are the first-stage results without controls, whereas specifications (2) and (4) report the results after controlling for the child's characteristics, mother's characteristics, household characteristics, and geographical location of households.

We see that the coefficient on the instrument is significant and positive in the specifications with controls, implying that the empowerment of the mother increases with the number of sons she gives birth. We see that this result holds for both indices. The results show that one extra son increases the mother's empowerment by 0.119 index points if it is measured by the additive index and by 0.00679 index points from the index generated by MCA.

The number of sons is a strong instrument only when appropriate variables are controlled for in the regression. Since the instrument is selected from within the household, it is only

**Table 4. Measuring the impact of total number of sons on the additive woman empowerment index and woman empowerment index measured by MCA.**

| Variables | Woman Empowerment (Additive Index) | | Woman Empowerment (MCA) | |
|---|---|---|---|---|
| | Without controls (1) | With Controls (2) | Without Controls (3) | With Controls (4) |
| Total Number of Sons | -0.178*** | 0.119*** | -0.032*** | 0.0067** |
| | (0.0504) | (0.0440) | (0.00280) | (0.0025) |
| Observations | 4,606 | 4,604 | 4,606 | 4,604 |
| F test | 12.52 | 7.32 | 129.42 | 5.55 |
| 1st P value | 0.0004 | 0.0069 | 0.0000 | 0.0186 |

Note: The two dependent variables are mothers' empowerment indices constructed in two different ways; the first is the additive index, and the second is the index created by multiple correspondence analysis (MCA). The instrument used in the first-stage regression is the total number of sons born to a mother in a household. Other controls include the child's characteristics: gender, age, and age squared; household characteristics: urban, gender of the household head, total number of households, wealth score, household head education level, wealth score square; mother's characteristics: mother's education level, mother's age, mother's age squared, age of the first born, number of years married. The geographical controls comprise district and province fixed effects. Standard errors are clustered at the household level

*** p<0.01

** p<0.05

* p<0.1

under a specific setting that it becomes exogenous. The number of sons on its own might not pass the orthogonality condition, but if we control for other household characteristics, the number of sons become positively significant and exogenously determined as shown in column (4). Therefore, only controlling for number of sons is not sufficient and first-stage results in column (3) are biased. However, to prove our point we use the Hausman Test for endogeneity and show that the instrument passes the exogeneity criteria (S4 Table).

**3.1.2 Second-stage results.** We report the second-stage results in Table 5 that measure the impact of the mother's empowerment (measured by both indices) on the health outcomes measured by the two anthropometric measures HFA and WFA, controlling for the child's characteristics, mother's characteristics, household characteristics, and geographical location.

The results show a positive and significant effect of greater empowerment of mothers on the short-term measure of health outcome for younger children, i.e., WFA. We report the OLS results for each respective measure of health outcome in specifications (1), (2), (5) and (6). Specification (4) using the additive index shows that the weight-for-age of children increases by 0.298 standard deviations (SD) (significant at the 5% significance level) if the empowerment index improves by 1 index point. Similarly, specification (8) shows that as we use the MCA index to measure the empowerment of the mother, the weight-for-age of younger children increases 1.168 SD for a unit increase in the mother's empowerment index. We do not find any significant increases in the value of HFA, which is considered a long-term measure of health outcomes.

The results show that empowered mothers are crucial for the improvement of a child's health. Empowered mothers become capable of improving the health outcomes of their children due to the choices they may take differently as compared to the disempowered mothers. From providing more nutritious food to utilizing better health services and observing improved hygiene and sanitation, empowered mothers can directly advocate for their child's health and wellbeing and to take greater precautions.

Next, we divide the sample into rural and urban regions. Table 6 reports the estimation of the impact of greater empowerment of mothers on the health outcomes of younger children

**Table 5. Measuring the impact of mother's empowerment on the child's health outcomes, on average.**

| Dependent Variable | Empowerment measured Using Additive Index | | | | Empowerment measured Using MCA | | | |
|---|---|---|---|---|---|---|---|---|
| | Weight for age Z-Scores | | | | | | | |
| | OLS (1) | OLS with controls (2) | IV (3) | IV with controls (4) | OLS (5) | OLS with controls (6) | IV (7) | IV with controls (8) |
| Empowered Mother | 0.0331*** | 0.00386 | 0.184* | 0.298** | 0.120*** | 0.011 | 0.622* | 1.168* |
| | (0.00690) | (0.00734) | (0.101) | (0.145) | (0.024) | (0.026) | (0.338) | (0.611) |
| **Observations** | 4,606 | 4,604 | 4,606 | 4,604 | 4,606 | 4,604 | 4,606 | 4,604 |
| 1st partial R2 | 0.012 | 0.275 | | | 0.013 | 0.275 | | |
| 1st Stage F Value | | | 23.089 | 6.226 | | | 24.674 | 6.226 |
| Dependent Variable | Height for age Z-Scores | | | | | | | |
| Empowered Mother | 0.0165*** | 0.00557 | 0.145 | 0.181 | 0.060*** | 0.018 | 0.488 | 0.711 |
| | (0.00589) | (0.00752) | (0.102) | (0.133) | (0.021) | (0.026) | (0.340) | (0.549) |
| Observations | 5,158 | 5,156 | 5,158 | 5,156 | 5,158 | 5,156 | 5,158 | 5,156 |
| 1st partial R2 | 0.003 | 0.123 | | | 0.003 | 0.123 | | |
| 1st Stage F Value | | | 7.835 | 3.282 | | | 8.428 | 3.282 |

Note: The two dependent variables are height-for-age z-scores and weight-for-age z-scores for the children of age group 5 years and less. The main independent variable, mother's empowerment is measured by indices constructed in two different ways: first, the additive index, and second, the index created by the multiple correspondence analysis (MCA). Controls include the child's characteristics: gender, age, and age squared. Household characteristics: urban, gender of the household head, total number of households, wealth score, household head education level, wealth score square. Mother's characteristics: mother's education level, mother's age, mother's age squared, age of the first born, number of years married. The geographical controls comprise district and province fixed effects. Standard errors are clustered at the household level

*** p<0.01

** p<0.05

* p<0.1

based on whether they live in rural or urban areas. The results show that household location plays an important role in the impact of empowerment of mothers. We see that a 1 unit increase in the additive empowerment index results in a 0.622 SD increase in WFA for children in rural areas alone. Similarly, when using the MCA index, a unit increase in the index of mothers' empowerment leads to a 1.33 SD increase in the weight-for-age of younger children in rural areas.

We argue that under constrained resources the impact of empowered mothers become more pronounced. With limited access to health care services and poor education about health care and nutritional intake, empowered mothers may be able to mitigate the impacts of these constraints on their child's health and can find ways to combat them.

Next, we test this relationship for girls and boys separately, and the results are shown in Table 7.

We see that the results for the combined sample (given in Table 5) are driven by the results for girls. We find that a 1 unit increase in the additive index leads to an increase in the WFA z-scores by 0.401 SD that is not significant, while a 1 unit increase in the MCA index leads to an insignificant significant increase in WFA z-score by 1.576 SD for girls. However, we see a significant increase in the long-term measure of health outcomes i.e., HFA z-Scores. The results show that a one unit increase in the additive index measuring mother's empowerment increase the HFA z-scores by 0.260 SD and 0.961 SD if we measure mother's empowerment by MCA (both significant at 10% significance level).

These results imply that child health outcomes are not only the result of better nutritional intake and improved hygiene and sanitation but also the result of empowering mothers. This

**Table 6. Measuring the impact of the mother's empowerment on the child's health outcomes by rural/urban divide.**

| | Empowerment measured using Additive Index | | | | Empowerment measured using MCA | | | |
|---|---|---|---|---|---|---|---|---|
| | OLS (1) | OLS with Controls (2) | IV (3) | IV with Controls (4) | OLS (1) | OLS with Controls (2) | IV (3) | IV with Controls (4) |
| Dependent variable | Weight for age Z-Scores | | | | | | | |
| Mother's empowerment Index | 0.017** | -0.019** | 0.188 | 0.622** | 0.047 | -0.071** | 4.418 | 1.333** |
| | (0.008) | (0.009) | (0.140) | (0.261) | (0.033) | (0.031) | (140.546) | (-0.091) |
| empowerment *urban | 0.028*** | 0.018 | -0.003 | -0.374 | 0.171*** | 0.074* | -6.128 | -1.347 |
| | (0.007) | (0.013) | (0.029) | (0.392) | (0.048) | (0.045) | (225.460) | (1.347) |
| Observations | 4,606 | 4,604 | 4,606 | 4,604 | 4,606 | 4,604 | 4,606 | 4,604 |
| R-squared | 0.020 | 0.146 | | | 0.019 | 0.146 | | |
| 1st F-test | | | 18.601 | 24.161 | | | 20.089 | 24.289 |
| Dependent variable | Height for Age Z-Scores | | | | | | | |
| Mother's empowerment Index | 0.013* | -0.009 | 0.203 | 0.174 | 0.034 | -0.029 | -7.714 | 0.720 |
| | (0.007) | (0.008) | (0.150) | (0.202) | (0.028) | (0.029) | (29.239) | (0.741) |
| empowerment* urban | 0.005 | 0.008 | -0.035 | 0.065 | 0.061 | 0.025 | 11.647 | 0.122 |
| | (0.006) | (0.012) | (0.028) | (0.287) | (0.042) | (0.042) | (43.896) | (0.955) |
| Observations | 5,158 | 5,156 | 5,158 | 5,156 | 5,158 | 5,156 | 5,158 | 5,156 |
| R-squared | 0.003 | 0.041 | | | 0.004 | 0.041 | | |
| 1st F-test | | | 4.264 | 7.053 | | | 5.273 | 7.056 |

Note: The two dependent variables are height-for-age z-scores and weight-for-age z-scores for the children of age group 5 years and less. The main independent variable, mother's empowerment is measured by indices constructed in two different ways: first, the additive index, and second, the index created by the multiple correspondence analysis (MCA). Controls include the child's characteristics: gender, age, and age squared. Household characteristics: urban, gender of the household head, total number of households, wealth score, household head education level, wealth score square. Mother's characteristics: mother's education level, mother's age, mother's age squared, age of the first born, number of years married. The geographical controls comprise district and province fixed effects. Standard errors are clustered at the household level

\*\*\* $p < 0.01$

\*\* $p < 0.05$

\* $p < 0.1$

will not only improve the overall health outcomes of the children but may also encourage gender equality that can have far reaching impacts on society.

Finally, in Table 8 we divided the sample of households into wealthy and poor households. We define wealthy households as those that lie in the upper two wealth quintiles and poor households as those households that lie in the lower two wealth quintiles. In this case, we do not find significant differences in the impact based on the wealth of the households.

**3.1.3 Other control variables.** We can also see which other variables that we have controlled for affect health outcomes for children in Pakistan (S1 Table). We see that on average younger children tend to be significantly less healthy and health outcomes improve, as children get older. Similarly, we see that the health of children is significantly poorer if they have a younger mother while child health outcomes improve significantly, as the age of mothers crosses a certain threshold. Similarly, mother's education has a significant positive impact on the health outcomes of the children. We see that on average the health outcomes of boys are significantly worse than that of girls. Working women, on average tend to have children with significantly lower health outcomes. Likewise, we see that children in lower wealth quintile households have significantly lower health outcomes than children born in households from the top wealth quintile.

**Table 7. Measuring the impact of mother's empowerment on the child's health outcomes by gender.**

| | Empowerment measured using Additive Index | | | | Empowerment measured using MCA | | | |
|---|---|---|---|---|---|---|---|---|
| | OLS (1) | OLS with Controls (2) | IV (3) | IV with Controls (4) | OLS (1) | OLS with Controls (2) | IV (3) | IV with Controls (4) |
| Dependent Variable | Weight for Age Z-Scores | | | | | | | |
| Empowerment Additive Index | 0.031*** | -0.008 | 0.187** | 0.401 | 0.132*** | -0.031 | 0.602 | 1.576 |
| | (0.007) | (0.008) | (0.090) | (0.294) | (0.028) | (0.029) | (1.394) | (1.210) |
| empowerment*boy | 0.004 | -0.007 | 0.001 | 0.110 | -0.025 | -0.023 | 0.056 | 0.386 |
| | (0.004) | (0.008) | (0.008) | (0.190) | (0.031) | (0.029) | (3.138) | (0.682) |
| Observations | 4,606 | 4,604 | 4,606 | 4,604 | 4,606 | 4,604 | 4,606 | 4,604 |
| R-squared | 0.012 | 0.145 | | | 0.013 | 0.145 | | |
| 1st F-test | | | 11.988 | 24.183 | | | 12.938 | 24.168 |
| Dependent Variable | Height for age Z-Scores | | | | | | | |
| Empowerment Additive Index | 0.020*** | -0.007 | 0.119 | 0.260* | 0.060** | -0.027 | 1.002 | 0.961* |
| | (0.006) | (0.008) | (0.085) | (0.152) | (0.026) | (0.027) | (0.861) | (-0.562) |
| empowerment*boy | -0.007 | 0.002 | -0.006 | -0.169 | -0.001 | 0.014 | -1.443 | -0.569 |
| | (0.004) | (0.008) | (0.007) | (0.130) | (0.029) | (0.029) | (1.870) | (0.441) |
| Observations | 5,158 | 5,156 | 5,158 | 5,156 | 5,158 | 5,156 | 5,158 | 5,156 |
| R-squared | 0.003 | 0.041 | | | 0.003 | 0.041 | | |
| 1st F-test | | | 5.042 | 7.068 | | | 4.213 | 7.098 |

Note: The two dependent variables are height-for-age z-scores and weight-for-age z-scores for the children of age group 5 years and less. The main independent variable, mother's empowerment is measured by indices constructed in two different ways: first, the additive index, and second, the index created by the multiple correspondence analysis (MCA). Controls include the child's characteristics: gender, age, and age squared. Household characteristics: urban, gender of the household head, total number of households, wealth score, household head education level, wealth score square. Mother's characteristics: mother's education level, mother's age, mother's age squared, age of the first born, number of years married. The geographical controls comprise district and province fixed effects. Standard errors are clustered at the household level

*** $p < 0.01$

** $p < 0.05$

* $p < 0.1$.

## 3.2 Measuring the impact of the presence of grandmothers on the child's health outcomes

In this section, we present the results for the relationship between the presence of grandparents on two anthropometric measures of health, height-for-age and weight-for-age, for children of age group 5 years and less. First, we present the first-stage results followed by the second-stage results for the entire sample. In addition, to explore heterogeneity in the impact of the presence of grandmothers, we see if the relationship is different for households in rural areas as opposed to households in urban areas, if the relationship is different for girls as compared to boys and if the relationship is different for high-wealth households as compared to low-wealth households. The methodology for constructing wealth indices in the Multiple Cluster Survey (MICS) is similar to that used in PDHS. Though the variables used to construct the index using PCA in different rounds of survey might differ slightly, the overall exercise ensures that the households are divided into quintiles based upon their wealth resources. The households from the lowest two quintiles are categorized as constrained and severely constrained households, respectively.

**3.2.1 First-stage results.** Table 9 presents the first-stage results. In this, we measure the impact of the potential retirement eligibility criteria on the probability of the presence of at least one grandparent, the presence of only a grandmother and the presence of only a grandfather in a household to create exogenous variation in this decision at the household level.

**Table 8. Measuring the impact of mother's empowerment on the child's health outcomes by wealth.**

| | Empowerment measured using Additive Index | | | | Empowerment measured using MCA | | | |
|---|---|---|---|---|---|---|---|---|
| | OLS (1) | OLS with Controls (2) | IV (3) | IV with Controls (4) | OLS (1) | OLS with Controls (2) | IV (3) | IV with Controls (4) |
| Dependent variable | Weight for age Z-Scores | | | | | | | |
| Mother's empowerment Index | -0.013 | -0.017* | -0.004 | 0.543 | 0.048 | -0.057 | 7.801 | 1.128 |
| | (0.009) | (0.010) | (0.143) | (0.391) | (0.036) | (0.035) | (10.897) | (0.476) |
| Empowerment*rich | 0.075*** | 0.012 | 0.084*** | -0.198 | 0.158*** | 0.034 | -10.375 | -0.857 |
| | (0.007) | (0.013) | (0.028) | (0.568) | (0.049) | (0.046) | (17.920) | (1.959) |
| Observations | 4,606 | 4,604 | 4,606 | 4,604 | 4,606 | 4,604 | 4,606 | 4,604 |
| R-squared | 0.068 | 0.145 | 0.065 | | 0.018 | 0.145 | | |
| 1st F-test | | | 77.950 | 23.914 | | | 21.706 | 23.918 |
| Dependent Variable | Height for age Z-Scores | | | | | | | |
| | | | | | | | | |
| Mother's empowerment Index | -0.010 | -0.007 | 0.041 | 0.385 | 0.025 | -0.024 | 25.047 | 1.443 |
| | (0.007) | (0.009) | (0.183) | (0.466) | (0.029) | (0.030) | (402.348) | (1.588) |
| Empowerment*rich | 0.042*** | 0.004 | 0.042 | -0.274 | 0.076* | 0.010 | -38.201 | -1.029 |
| | (0.006) | (0.012) | (0.039) | (0.751) | (0.043) | (0.042) | (634.749) | (2.630) |
| Observations | 5,158 | 5,156 | 5,158 | 5,156 | 5,158 | 5,156 | 5,158 | 5,156 |
| R-squared | 0.021 | 0.041 | | | 0.004 | 0.041 | | |
| 1st F-test | | | 27.090 | 7.045 | | | 5.734 | 7.043 |

Note: The two dependent variables are height-for-age z-scores and weight-for-age z-scores for the children of age group 5 years and less. The main independent variable, mother's empowerment is measured by indices constructed in two different ways: first, the additive index, and second, the index created by the multiple correspondence analysis (MCA). Controls include the child's characteristics: gender, age, and age squared. Household characteristics: urban, gender of the household head, total number of households, wealth score, household head education level, wealth score square. Mother's characteristics: mother's education level, mother's age, mother's age squared, age of the first born, number of years married. The geographical controls comprise district and province fixed effects. Standard errors are clustered at the household level

*** $p < 0.01$

** $p < 0.05$

* $p < 0.1$

We choose different age cutoffs for grandmothers and grandfathers to implement the fuzzy regression discontinuity design. The cutoff age for grandmothers is kept at 55 years, whereas the age cutoff of grandfathers is kept at 60 years of age based upon the rationale that the legal retirement age of females is different from that of males. We find that the potential retirement eligibility criteria (age minus the potential retirement age) has a significant and positive impact on the respective probabilities of the presence of at least one grandparent, the presence of only grandfathers and the presence of only grandmothers. The results show that the probability of the presence of grandparents in a household increases by 2.91 percentage points as the age of the grandparents increases by one extra year above the retirement eligibility criteria (i.e., 55 years for grandmothers and 60 years for grandfathers). The probability of the presence of a grandfather in a household increase by 3.27 percentage points if the age of the grandfather increases by one additional year above 60 years. Similarly, one extra year above 55 years (retirement eligibility criteria for women) significantly affects the probability of the presence of grandmothers in a household by 3.58 percentage points. In S3 Table, we present results using different retirement age cutoffs for men and women and find that the best results are obtained in the case of a retirement age of 55 years for women and 60 years for men. This shows that our results are robust to the specific age cut-off selected in the main analysis.

**Table 9. First stage results: Measuring the impact of retirement eligibility criteria on the availability of grandparents (both or at least one), grandfather (alone) and grandmother (alone) in an HH.**

| Dependent Variable: Actual age -Retirement Eligibility Criteria | Without Controls (1) | With Controls (2) | Without Controls (3) | With Controls (4) | Without Controls (5) | With Controls (6) |
|---|---|---|---|---|---|---|
| Dummy = 1 if the household has at least one Grandparent | 0.0282*** | 0.0291*** | | | | |
| | (0.000264) | (0.000360) | | | | |
| Dummy = 1 if household has only Grandfather | | | 0.0329*** | 0.0327*** | | |
| | | | (0.000319) | (0.000431) | | |
| Dummy = 1 if the household has only Grandmother | | | | | 0.0353*** | 0.0358*** |
| | | | | | (0.000143) | (0.000197) |
| Observations | 187,918 | 99,218 | 187,918 | 99,218 | 187,918 | 99,218 |
| F test | 11407.7 | 6504.46 | 6504.7 | 5740.3 | 60510.7 | 33016.35 |
| 1st P value | 0.00 | 0.00 | 0.00 | 0.00 | 0.00 | 0.00 |

Note: The dependent variables comprise a dummy variable that takes a value of 1 if both grandparents are present in specifications (1), (2) and (3). Dummy = 1 if only grandfather is present in the household is used in specification (4) and dummy = 1 if only grandmother is present in the household in specification (5). Instruments used in the specification are RE, which is grandparent's age minus the legal retirement age (60 years for males and 55 years for females). The controls include the child's characteristics: gender, age, and age squared. Household characteristics: urban, gender of the household head, total number of households, wealth score, household head education level, wealth score square. Mother's characteristics: mother's education level, mother's age, mother's age squared, age of the first born, number of years married, district fixed effects, year fixed effects, mother's age at the birth of child, dummy = 1 if there are 2 children and above in a household, dummy = 1 if there are 3 children and above in a household, dummy = 1 if the second and above child is a girl, dummy = 1 if the third and above child is a girl. Standard errors are clustered at the household level

*** p<0.01

** p<0.05

* p<0.1

**3.2.2 Second-stage results.** We report the second-stage results that measure the impact of the presence of at least one grandparent, only a grandfather, and only a grandmother on the nutritional health of younger children, as measured by WFA and HFA, in Table 10.

The OLS results reported in columns (1) and (2) show that there is a positive and significant effect of the presence of at least one grandparent in a household on the height-for-age and weight-for-age measures. The IV results reported in column (3) show that the weight-for-age increases by 0.0862 SD if at least one grandparent is present in a household, and similarly, the height-for-age also increases by 0.101 SD if at least one grandparent is present in a household. We see that the results remain insignificant for the households that only have grandfathers (column (4)). The final specification column (5) shows that the results of the grandparents are driven by the presence of grandmothers. The coefficient for weight-for-age in the specification that captures the impact of the presence of grandmothers (alone) in a household is more significant and larger in magnitude, i.e., 0.0984 SD then that obtained when we look at the presence of at least one grandparent. HFA, on the other hand, remains positive but insignificant.

The results show that the presence of a grandmother plays a positive role in the lives of children in a household. These results can be attributed to the support that the grandmothers may provide to the children. Grandmothers can pass down traditional methods and different perspective for the care of children and can also pass on important lessons learned to household members. In addition, they can be a close substitute to the parents for short time periods that can help the parents support in childcare that not only increases the overall level of care but also reduces the burden on parents.

**Table 10. Measuring the impact of the presence of grandparents on the child's health outcomes, on average.**

| | OLS (1) | OLS with Controls (2) | IV with Controls (3) | IV with Controls (4) | IV with Controls (5) |
|---|---|---|---|---|---|
| Dependent variable | Weight for Age Z-Scores | | | | |
| Dummy = 1 if the household has at least one Grandparent | 0.115*** | 0.0264** | 0.0862* | | |
| | (0.00853) | (0.0113) | (0.0455) | | |
| Dummy = 1 if household has only Grandfather | | | | 0.0567 | |
| | | | | (0.0541) | |
| Dummy = 1 if the household has only Grandmother | | | | | 0.0984** |
| | | | | | (0.0451) |
| Observations | 187,918 | 99,218 | 99,218 | 99,218 | 99,218 |
| 1st partial R2 | 0.001 | 0.114 | 0.114 | 0.114 | 0.114 |
| Dependent variable | Height for Age Z-Scores | | | | |
| Dummy = 1 if the household has at least one Grandparent | 0.146*** | 0.0405*** | 0.101* | | |
| | (0.0109) | (0.0137) | (0.0553) | | |
| Dummy = 1 if household has only Grandfather | | | | 0.0813 | |
| | | | | (0.0658) | |
| Dummy = 1 if the household has only Grandmother | | | | | 0.0823 |
| | | | | | (0.0548) |
| Observations | 184,124 | 98,229 | 98,229 | 98,229 | 98,229 |
| 1st partial R2 | | | 0.125 | 0.125 | 0.125 |

Note: The dependent variables are weight-for-age (WFA) and height-for-age (HFA). The independent variable comprises the main dummy variable, which takes a value of 1 if the grandmother is present in a household. Specifications (1) and (2) report OLS results, whereas specifications (3) and (3) report the IV results. The instrument used in the specification is Retirement Eligibility, which is grandmother's age minus the potential retirement age (55 years for females). The controls include the child's characteristics: gender, age, and age squared. Household characteristics: urban, gender of the household head, total number of households, wealth score, household head education level, wealth score square. Mother's characteristics: mother's education level, mother's age, mother's age squared, age of the first born, number of years married, district fixed effects, year fixed effects, Mothers age at the birth of child, dummy = 1 if there are 2 children and above in a household, dummy = 1 if there are 3 children and above in a household, dummy = 1 if the second and above child is a girl, dummy = 1 if the third and above child is a girl. Standard errors are clustered at the household level

*** p<0.01

** p<0.05

* p<0.1

Fig 4 shows the regression discontinuity around the retirement eligibility criteria. The weight for the age z-score of children (age group 5 years and less) is on the y-axis, and the actual age of the grandmother minus the retirement eligibility age (55 years) is on the x-axis.

Next, we subdivide our sample on the basis of location, gender of children, and wealth of households. Table 11 reports the results for the impact of the presence of grandmothers on the HFA and WFA of child outcomes in rural and urban households.

We see that the results for WFA are driven by households living in rural areas. The weight-for-age of younger children increases by 0.160 SD if only grandmothers are present in rural households. However, we do not see any significant increases in HFA. The importance of grandmothers in rural areas may become more pronounced due to the limited resources available to the household. In such conditions, grandmothers can help in the upbringing of the children and can provide financial and emotional support to them. Furthermore, they can serve as a strong role model that can pass traditional information and knowledge about childcare practices to improve the health outcomes of children.

Next, we look to see if the impact of grandmothers is different for boys versus girls. Table 12 shows that the positive impact of the presence of grandmothers on WFA is driven by

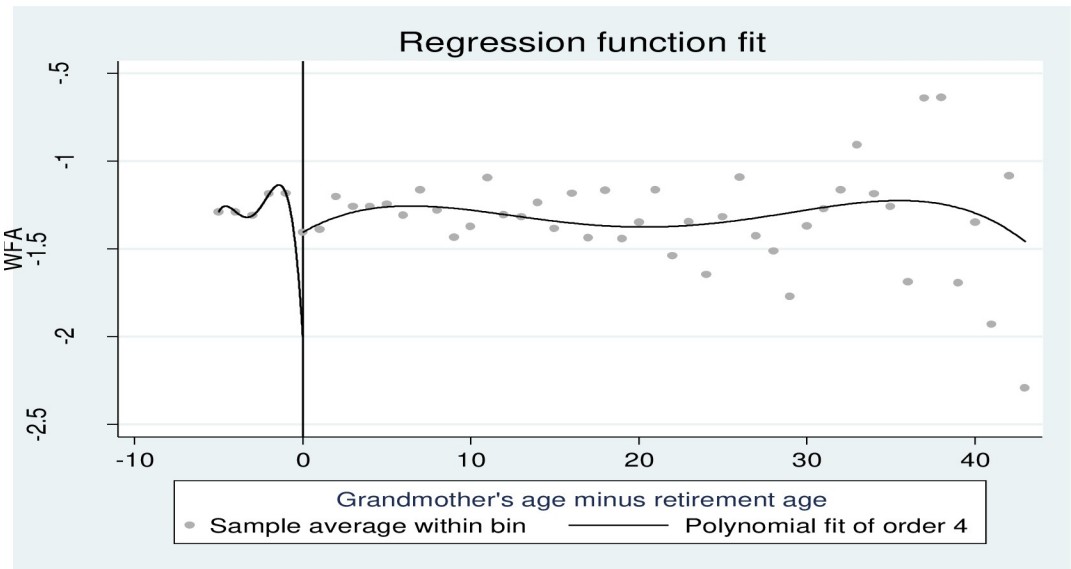

**Fig 4. Regression discontinuity design for WFA at the grandmother's age cutoff at 55 years.**

**Table 11. Measuring the impact of the presence of grandmothers in a household on children's health outcomes by rural/urban divide.**

| | OLS (1) | OLS with Controls (2) | IV (3) | IV with Controls (4) |
|---|---|---|---|---|
| Dependent Variable | Weight for Age Z-Scores | | | |
| Dummy = 1 if the households have only Grandmother | 0.003 | -0.004 | 0.054 | 0.160*** |
| | (0.022) | (0.028) | (0.042) | (0.057) |
| Presence of Grandmother only * Urban | 0.210*** | -0.054 | 0.241*** | -0.178* |
| | (0.038) | (0.047) | (0.074) | (0.091) |
| Observations | 187,918 | 99,218 | 187,918 | 99,218 |
| R-squared | 0.000 | 0.114 | | |
| 1st F-test | | | 24.566 | 238.396 |
| Dependent Variable | Height for Age Z-Scores | | | |
| Dummy = 1 if the households have only Grandmother | 0.004 | -0.023 | 0.019 | 0.079 |
| | (0.027) | (0.035) | (0.051) | (0.068) |
| Presence of Grandmother only * Urban | 0.263*** | 0.001 | 0.399*** | 0.010 |
| | (0.046) | (0.054) | (0.091) | (0.108) |
| Observations | 184,124 | 98,229 | 184,124 | 98,229 |
| R-squared | 0.000 | 0.125 | | |
| 1st F-test | | | 25.771 | 292.298 |

Note: The dependent variables are weight-for-age (WFA) and height-for-age (HFA). The independent variable comprises the main dummy variable, which takes a value of 1 if the grandmother is present in a household. Specifications (1) and (2) report OLS results, whereas specifications (3) and (3) report the IV results. The instrument used in the specification is Retirement Eligibility, which is grandmother's age minus the potential retirement age (55 years for females). The controls include the child's characteristics: gender, age, and age squared. Household characteristics: urban, gender of the household head, total number of households, wealth score, household head education level, wealth score square. Mother's characteristics: mother's education level, mother's age, mother's age squared, age of the first born, number of years married, district fixed effects, year fixed effects, Mothers age at the birth of child, dummy = 1 if there are 2 children and above in a household, dummy = 1 if there are 3 children and above in a household, dummy = 1 if the second and above child is a girl, dummy = 1 if the third and above child is a girl. Standard errors are clustered at the household level

*** p<0.01

** p<0.05

* p<0.1

**Table 12. Measuring the impact of the presence of grandmothers in a household on children's health outcomes by gender.**

| | OLS (1) | OLS with Controls (2) | IV (3) | IV with Controls (4) |
|---|---|---|---|---|
| Dependent Variable | Weight for Age Z-Scores | | | |
| Dummy = 1 if the households have only Grandmother | 0.070*** | -0.030 | 0.137*** | 0.169** |
| | (0.024) | (0.032) | (0.048) | (0.067) |
| Presence of Grandmother only * Girl | 0.019 | 0.013 | -0.019 | -0.120 |
| | (0.034) | (0.043) | (0.068) | (0.088) |
| Observations | 187,918 | 99,218 | 187,918 | 99,218 |
| R-squared | 0.000 | 0.114 | | |
| 1st F-test | | | 9.919 | 238.170 |
| Dependent Variable | Height for Age Z-Scores | | | |
| Dummy = 1 if the households have only Grandmother | 0.071** | -0.047 | 0.152** | 0.099 |
| | (0.030) | (0.038) | (0.060) | (0.078) |
| Presence of Grandmother only * Girl | 0.055 | 0.051 | -0.023 | -0.033 |
| | (0.042) | (0.050) | (0.084) | (0.106) |
| Observations | 184,124 | 98,229 | 184,124 | 98,229 |
| R-squared | 0.000 | 0.125 | | |
| 1st F-test | | | 10.773 | 292.232 |

Note: The dependent variables are weight-for-age (WFA) and height-for-age (HFA). The independent variable comprises the main dummy variable, which takes a value of 1 if the grandmother is present in a household. Specifications (1) and (2) report OLS results, whereas specifications (3) and (3) report the IV results. The instrument used in the specification is Retirement Eligibility, which is grandmother's age minus the potential retirement age (55 years for females). The controls include the child's characteristics: gender, age, and age squared. Household characteristics: urban, gender of the household head, total number of households, wealth score, household head education level, wealth score square. Mother's characteristics: mother's education level, mother's age, mother's age squared, age of the first born, number of years married, district fixed effects, year fixed effects, Mothers age at the birth of child, dummy = 1 if there are 2 children and above in a household, dummy = 1 if there are 3 children and above in a household, dummy = 1 if the second and above child is a girl, dummy = 1 if the third and above child is a girl. Standard errors are clustered at the household level

*** p<0.01

** p<0.05

* p<0.1

the impact on boys. The weight-for-age of younger individuals increases by 0.169 SD for boys in the households if only the grandmothers are present. We do not see any significant effect on the HFA. The results show that although there is no negative impact of the presence of grandmother on the health outcomes of girls, we see a significantly positive impact on the health outcomes of the boys implying that on average the boys receive more benefits due to the presence of grandmothers as compared to girls. These results potentially indicate a certain amount of gender preference in households.

Finally, in Table 13, we see that the results for the increase in WFA and HFA due to the presence of grandmothers in a household occur primarily in low-wealth households. The weight-for-age of younger individuals increases by 0.184 SD in poor households if only the grandmother is present. Similarly, we see a significant increase in HFA by 0.163 SD in low-wealth households if the grandmother is present.

These results again reinforce the fact that the presence of grandmothers play a trivial role in households that are more vulnerable and constrained. Grandmother's presence can facilitate parents by offering childcare that allows them to potentially work outside of the home.

**3.2.3 Other controls.** We report the results of the entire specification measuring the impact of the other control variables in our analysis of the impact of grandmothers in a

**Table 13. Measuring the impact of the presence of grandmothers in a household on children's health outcomes by wealth.**

| | OLS (1) | OLS with Controls (2) | IV (3) | IV with Controls (4) |
|---|---|---|---|---|
| Dependent Variable | Weight for Age Z-Scores | | | |
| Dummy = 1 if the households have only Grandmother | -0.275*** | -0.004 | -0.280*** | 0.184** |
| | (0.030) | (0.041) | (0.056) | (0.075) |
| Presence of Grandmother only * Rich | 0.520*** | -0.029 | 0.644*** | -0.128 |
| | (0.037) | (0.049) | (0.070) | (0.093) |
| Observations | 187,918 | 99,218 | 187,918 | 99,218 |
| R-squared | 0.001 | 0.114 | 0.001 | 0.114 |
| 1st F-test | | | 107.736 | 238.174 |
| Dependent Variable | Height for Age Z-Scores | | | |
| Dummy = 1 if the households have only Grandmother | -0.326*** | -0.028 | -0.386*** | 0.163* |
| | (0.039) | (0.053) | (0.069) | (0.092) |
| Presence of Grandmother only * Rich | 0.624*** | 0.009 | 0.832*** | -0.122 |
| | (0.047) | (0.061) | (0.087) | (0.111) |
| Observations | 184,124 | 98,229 | 184,124 | 98,229 |
| R-squared | 0.001 | 0.125 | | |
| 1st F-test | | | 100.272 | 292.368 |

Note: The dependent variables are weight-for-age (WFA) and height-for-age (HFA). The independent variable comprises the main dummy variable, which takes a value of 1 if the grandmother is present in a household. Specifications (1), (3), (5) and (7) report the OLS results, whereas specifications (2), (4), (6) and (8) report the IV results. The instrument used in the specification is Retirement Eligibility, which is grandmother's age minus the potential retirement age (55 years for females). The controls include the child's characteristics: gender, age, and age squared. Household characteristics: urban, gender of the household head, total number of households, wealth score, household head education level, wealth score square. Mother's characteristics: mother's education level, mother's age, mother's age squared, age of the first born, number of years married, district fixed effects, year fixed effects, Mothers age at the birth of child, dummy = 1 if there are 2 children and above in a household, dummy = 1 if there are 3 children and above in a household, dummy = 1 if the second and above child is a girl, dummy = 1 if the third and above child is a girl. Standard errors are clustered at the household level

*** $p < 0.01$

** $p < 0.05$

* $p < 0.1$

household on the health outcomes of the children in S2 Table. We see similar impacts as that we have reported in section 3.1. The results show that on average girls are healthier than boys and that as the age of the child increases, they become healthier. Children in urban areas are healthier than the children located in rural areas. Children's health improves as the wealth of the household improves. Younger mothers tend to have children with lower health outcomes as compared to mothers above a threshold age.

## 4. Conclusion

We have analyzed how the household structure and female agency can alleviate the undernutrition of children, a severe problem faced by approximately 40% of children (of age group 5 years and less) in Pakistan (United Nations International Children's Emergency Fund (UNICEF), 2018). Our study is novel in that we quantitatively measure the causal impact of the two important household factors, the empowerment of mothers and the presence of grandparents, on the health outcomes of children under age-5 in the case of Pakistan.

Where empowered mothers provide benefits to girls, we see that the presence of grandparents, more specifically that of grandmothers, improves the health status of boys, children in poor families, and the children of families living in rural areas.

The literature has found that empowered mothers have a positive impact on child health outcomes. Different ways have been proposed in literature to measure the empowerment of mothers. For instance, using the size of the mother's social network to measure her empowerment, [33] showed for a small city in India that mothers with larger networks had access to a wider range of resources, and that had a positive effect on length-for-age z-scores (LAZ) of their children. However, our analysis revolves around the definition of empowerment based upon the authority of the women over decision-making process with in the household. For selective autonomy that a woman may have in a household, [12] uses a limited set of questions to measure empowerment and finds positive impacts of autonomy on stunting, wasting and child being underweight. He only incorporates the decision of the mother regarding the health care of children to measure her empowerment. Similarly, [34] reports positive impacts on the HFA z-scores and WFA z-scores if the mother is perceived to decide upon the number of children she wants to conceive. [3] reports that a mother has healthier children in terms of significantly higher HFA z-scores, WFA z-scores, and LFA z-scores if she is actively engaged in taking decisions regarding child care, cooking, and food supplies. Most authors used either a limited set of questions to measure the empowerment of mothers or used a simple linear regression framework to estimate the impact of empowerment on the child's health outcomes.

In this study, not only do we use an instrumental variable approach to correct for the endogenous empowerment of mothers in a household but also take into account larger set of questions (intrinsic and extrinsic) to create an index of mother's empowerment using PDHS 2017–18. This analysis not only shows an improvement in the short-term measure of health outcomes (WFA z-scores) for children in rural areas but also indicate that mothers' empowerment has ensured a long-term impact on girls' health outcome i.e., improvement in HFA z-scores. These results shed light on the longer term impact of mothers' empowerment on the next generation of women and therefore acting as a potential driver of positive change in future generations.

As far as the relevance of grandmothers is concerned, we see that the literature is divided when it comes to measuring the impact of co-resident grandparents, specifically grandmothers on the health outcomes of the children. While we see substantial differences in the degree of involvement of grandparents in lives of their children, it has been accepted widely that grandparents influence the lives of their grandchildren. One strand of the literature argues that over-indulgence and division of constrained resources due to the presence of grandmothers may have a negative impact on the health outcomes of the children [35, 36]. On the other hand, a large body of literature reports positive impacts of the presence of grandmothers on the health outcomes of the children due to two distinct reasons; first, providing informal child care and secondly, sharing their wealth of knowledge and other resources such as inheritance and social networks [37, 38]. Numerous studies have also reported obesity in the children living with grandmothers [39–41].

There are several caveats to interpreting the results associated with height for age z-scores and weight for age z-scores of children. First, Pakistan Demographic Health survey (PDHS) and Multiple Indicator Cluster Survey (MICS) use self-reported as well as direct measurement methods to collect information about anthropometric measures. Where direct measurement should be preferred, we accept that self-reported data collection can introduce measurement error, limited accuracy, and even low response. Second, the nutritional intake habits of different societies as well as genetic and ethnic diversity can create differences amongst the children across communities. In our analysis, we carefully introduce an exogenous variation in both the empowerment of mothers and the presence of grandmothers to interpret what could have happened if they were compared to their respective counterfactuals.

Many of these studies have used qualitative analysis and thematic analysis to estimate the impact of the presence of the grandmothers on child health outcomes. We use a fuzzy regression discontinuity design to alleviate the problem of the endogenous decision of grandmothers to be living within a household. Our results reinforce the importance of grandmothers in a south Asian country especially, where the resources are limited and their presence can provide an informal childcare. Not only do we show that on average the short-term measure of health improves in the households with grandmothers but we also show that larger gains are associated with children categorized as being in "vulnerable" groups, i.e. in rural areas and poorer households. However, we show that boys get significantly higher benefits from grandmothers as compared to girls.

This study statistically provides insight into the relevance of family structures for the betterment of children's health and sheds light on new aspects that can be explored in future research. A few important questions that arise from this research are the specific role played by empowered mothers and the presence of grandmothers in these households and how it affects the health of children. Eventually, in future studies, one can explore how the presence of grandmothers may affect the empowerment of mothers in a household and what is the combined impact of empowered mothers and the presence of grandmothers on the wellbeing of children.

## Supporting information

**S1 Table. Measuring the impact of mother's empowerment on the child nutritional outcomes using PDHS.**
(PDF)

**S2 Table. Measuring the impact of the presence of grandmother on the nutritional outcomes of the child in a household using MICS.**
(PDF)

**S3 Table. Robustness checks second-stage results with different cutoffs.**
(PDF)

**S4 Table. The results for the Hausman test.**
(PDF)

**S1 Fig. Distribution of height for age z-score for children age 5 and less using PDHS.**
(TIF)

**S2 Fig. Distribution of weight for age z-score for children age 5 and less using PDHS.**
(TIF)

**S3 Fig. Distribution of height for age z-score for children age 5 and less using MICS.**
(TIF)

**S4 Fig. Distribution of weight for age z-score for children age 5 and less using MICS.**
(TIF)

## Acknowledgments

We are grateful for the comments from Naved Hamid (Dean & Prof., Centre for Reasearch in Economics and Business, Lahore School of Economics), Aimal Tanvir (Senior Research Fellow, Lahore School of Economics and Fizza Naveed (Mphil, Lahore School of Economics) in the early stages of drafting the paper.

## Author Contributions

**Conceptualization:** Rabia Arif, Azam Chaudhry.

**Data curation:** Rabia Arif.

**Formal analysis:** Rabia Arif, Theresa Chaudhry.

**Investigation:** Rabia Arif.

**Methodology:** Rabia Arif.

**Supervision:** Azam Chaudhry, Theresa Chaudhry.

**Writing – original draft:** Rabia Arif.

**Writing – review & editing:** Azam Chaudhry, Theresa Chaudhry.

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
