## [Decision Letter · Decision Letter 0]

25 Nov 2022

PONE-D-22-19975Empowered Mothers & Coresident Paternal Grandmothers: Two Fundamental Factors Impacting Child Health Outcomes in Punjab, PakistanPLOS ONE

Dear Dr. Arif,

Thank you for submitting your manuscript to PLOS ONE. After careful consideration, we feel that it has merit but does not fully meet PLOS ONE’s publication criteria as it currently stands. Therefore, we invite you to submit a revised version of the manuscript that addresses the points raised during the review process.

Major Revisions

We look forward to receiving your revised manuscript.

Kind regards,

Faisal Abbas, PhD

Academic Editor

PLOS ONE

https://journals.plos.org/plosone/s/fileid=ba62/PLOSOne_formatting_sample_title_authors_affiliations.pdf.

2. PLOS ONE does not copy edit accepted manuscripts (https://journals.plos.org/plosone/s/criteria-for-publication#loc-5). To that effect, please ensure that your submission is free of typos and grammatical errors, including the equations.

Additional Editor Comments:

Major Revisions.

Reviewers' comments:

Reviewer's Responses to Questions

**Comments to the Author**

1. Is the manuscript technically sound, and do the data support the conclusions?

Reviewer #1: Yes

Reviewer #2: Yes

2. Has the statistical analysis been performed appropriately and rigorously? 

Reviewer #1: Yes

Reviewer #2: Yes

3. Have the authors made all data underlying the findings in their manuscript fully available?

Reviewer #1: Yes

Reviewer #2: Yes

4. Is the manuscript presented in an intelligible fashion and written in standard English?

Reviewer #1: Yes

Reviewer #2: Yes

5. Review Comments to the Author

Reviewer #1: The authors have estimated the effect of women empowerment on child health outcomes by using different waves of MICS data and further they have estimated the effect of co-residing grandparents on the child health outcome by using PDHS data for Punjab. I have thoroughly enjoyed the work and the effort which the authors have put to answer such important questions but I have some observations and suggestions which I believe will improve the quality of the work.

One of the main concerns is regarding the use of PCA because from provided details in the data section it seems that all the variables concerning women empowerment are discrete. It is possible to use PCA on discrete or even categorical variables that have one hot encoded variable but one must avoid it. In other words, if the variables do not belong to the coordinate plane, you should avoid using PCA. The only way PCA is a valid method of feature selection is if the most important variables are the ones that happen to have the most variation in them which is not possible with discrete or even categorical variables. If my concern is right and you have used PCA on discrete variables concerning women empowerment I would suggest dropping PCA and using the additive index only. I am also open to a strong rebuttal if you think my observation is not valid.

Secondly, the authors should be upfront in explaining the anthropometric factors and grouping which drive the differences such as dietary intake, and ethnicity/race, and should further explain the caveats while relying on self-reported data such as highest and weight. Respondents can create systematic bias while providing such data. This can create an impact on the estimates, therefore, you should write about it, to make your reader aware of the possible shortcomings.

Thirdly, in the data section, I think the authors should make the readers aware of the methodology of data collection concerning MICS and PDHS. Even if both are population representatives of Punjab, still there is a possibility that the variation in the outcome variable is solely due to the assumption taken over the data collection methodology. Although, the idea is not to make a comparison because both samples are different in a way as one estimates the effect of women empowerment on child health outcomes and the other is for co-residing grandparents affecting child health outcomes but still there should be a section which explicitly discusses the differences in data collection methodologies in both datasets.

Lastly, one minor observation is, the study used three different samples, (i) measuring the impact of having at least one grandparent in the household, (i) measuring the impact of having only a grandfather in the household, and (iii) measuring the impact of having only the grandmother in the household on the health outcomes of children in a household but the title of the paper only reflects the paternal grandmothers?

Reviewer #2: 1. In the abstract it is a good idea to include some numbers from the result section to indicate the effect size of mother empowerment and presence of grandmothers with respective confidence intervals (p-values).

2. In the introduction section first paragraph apart from the global statistics of stunting and wasting, it will be more valuable if such numbers are also given for Pakistan and specifically Punjab to get a good idea about the current situation.

3. “The mother’s empowerment index constructed in two different ways: first, we use the sum of positive responses given, and second, we use a principal component index (PCA) for ten survey questions measuring the behavioral and attitudinal dimensions of daily choices mothers make within the household”. Why have you used two different measures for mother’s empowerment given same questions are used for both. What is the justification for using additive approach, will it be not biased as some factors are more important than others to measure empowerment. Secondly, after running the principal component analysis (PCA), which criteria was followed in context of the eigen values? What percentage of variation was covered by the principal component?

4. PDHS survey was conducted in 2017-18, please write the years in this format rather than writing 2018 alone.

5. “We use data from the household surveys conducted in Punjab in Pakistan (MICS has also collected information about two other provinces in Pakistan, Sindh and Baluchistan. In the case of Baluchistan, the quality of data is poor, especially with regards to the nutrition indicators, and therefore not much useful information can be extracted from those surveys.)”. This explains why Baluchistan was not selected for analysis but no reason is given for Sindh? However, it would be interesting if difference between Punjab and Sindh is observed for the role of grandparents given MICS collect reliable data for Sindh.

6. “Where ℎ is the dependent variable that comprises the nutritional health for children aged 5 years and younger”. However, there is no explanation given what the subscript g,h and i means for the dependent and independent variables. Please include the level which it is representing.

7. What is the reason behind using square term for the child age, mother’s age, household head age and wealth index in addition to their level form? There is no interpretation given of the coefficient of these square terms and what does its results signify in this manuscript?

8. “Therefore, to address this problem, we use the number of sons a woman has as an instrument for mother’s empowerment. We argue that the number of sons a woman has is correlated with the empowerment of a woman, especially in the context of South Asia, where sons are given special importance in a household but are orthogonal to the health outcomes of children aged 5 years and less in a household”. There is no backing of literature that number of sons is a good instrument for mother’s empowerment and how it is not correlated directly to health outcomes of children aged 5 years and less. Number of sons can also have a direct impact on the health outcome of children, parents might be more concerned about the health of boys and provide them better nutrients and health care as there exists a preference of boys in the South Asian countries. The theoretical framework is lacking for the choice of instrument in the manuscript. Please add the relevant literature to support the choice of this IV which should justify it.

9. What kind of wealth index is used in this study? Is it the DHS wealth index already developed in the dataset or some other type of wealth index is used? Please specify it in the manuscript. Also, for the MICS data set, which wealth index is used?

10. “Dummy=1 if the mother ever used contraceptives in table 1 b”. This variable is showing the minimum and maximum value of 0 and 8. However, given this is a binary variable the maximum should be equal to 1. Can you please clarify on this?

11. In table 1b, the minimum value of mother’s age is 15 and the maximum value for number of years married is 43. These minimum and maximum values are not looking realistic, please into it and provide explanation for it.

12. Apart from empowered mother and presence of grandparents, results of other independent variables are not reported and interpreted in the manuscript. Some of these independent variables are of interest which should be included in the manuscript for completeness of results. For example, given the empowerment level of mother, what is the effect of household head’s education level on the children health outcome?

13. Why separate regressions are estimated for region (urban/rural), gender (boys/girls) and wealth status (poor/wealthy)? Why interaction terms were not added with empowered mother and presence of grandparents to analyze the effect of region, gender and wealth with the impact variable (empowered mother and presence of grandparents)? Split regression does not account for heterogenous effect accurately as the significance depends on other covariates as well.

14. In section 3.1.3, the household are divided into wealthy and poor households. Can you please elaborate on why bottom two quintiles are considered as poor and upper two quintile are categorized as wealthy household? What about the third quintile which lies in between the upper and lower two quintiles? If the DHS wealth index is considered in this study, then it is a relative measure of wealth calculation according to the given sample. The households which are falling in the bottom two quintiles might not be poor is absolute terms as the DHS wealth index divides the whole sample into 5 equal quintiles. How would this effect the categorization of household as poor or wealthy?

15. There is no discussion section present in the manuscript which analyzes the results, provide its explanation and compare it with the literature. The results section is only interpretation the coefficients of the estimated models. Please add a discussion section in the manuscript where results are discussed and logical reasoning is given for the derived results while comparing it with international and national studies and recommending actions for better health outcomes of the children aged 5 years and below.

6. PLOS authors have the option to publish the peer review history of their article (what does this mean?). If published, this will include your full peer review and any attached files.

Reviewer #1: **Yes: **SYED HASSAN RAZA

Reviewer #2: **Yes: **Taimoor Ahmad

---

## [Author Response · Author response to Decision Letter 0]

22 Feb 2023

PONE-D-22-19975

Empowered Mothers & Coresident Paternal Grandmothers: Two Fundamental Factors Impacting Child Health Outcomes in Punjab, Pakistan

PLOS ONE

Response to Reviewers

Dear Dr. Faisal,

Thank you for giving us the opportunity to submit a revised draft of the manuscript titled, “Empowered Mothers & Coresident Paternal Grandmothers: Two Fundamental Factors Impacting Child Health Outcomes in Punjab, Pakistan” that has now been changed to “Empowered Mothers and Co-Resident Grandmothers: Two Fundamental Roles of Women Impacting Child Health Outcomes in Punjab, Pakistan” for publication in PLOS one. We are grateful to the editor and the reviewers for their valuable comments that has strengthened the paper. We have attempted to incorporate all of the suggestions given by the reviewers.

Please see below our responses to the reviewers’ comments. 

 Review Comments to the Author

Reviewer #1: The authors have estimated the effect of women empowerment on child health outcomes by using different waves of MICS data and further they have estimated the effect of co-residing grandparents on the child health outcome by using PDHS data for Punjab. I have thoroughly enjoyed the work and the effort which the authors have put to answer such important questions but I have some observations and suggestions which I believe will improve the quality of the work.

1. One of the main concerns is regarding the use of PCA because from provided details in the data section it seems that all the variables concerning women empowerment are discrete. It is possible to use PCA on discrete or even categorical variables that have one hot encoded variable but one must avoid it. In other words, if the variables do not belong to the coordinate plane, you should avoid using PCA. The only way PCA is a valid method of feature selection is if the most important variables are the ones that happen to have the most variation in them which is not possible with discrete or even categorical variables. If my concern is right and you have used PCA on discrete variables concerning women empowerment I would suggest dropping PCA and using the additive index only. I am also open to a strong rebuttal if you think my observation is not valid.

Author’s response: Thank you for raising a valid point. We have re-estimated the coefficients using Multiple Correspondence analysis (MCA) as proposed in literature to be used while handling qualitative information. In literature, researchers have used Principal Component Analysis as one of the potential ways to reduce the dimensionality of the research question and create indices based upon multiple questions; hence, it was used in the analysis earlier. However, the reviewers correctly identified that PCA is used for continuous variables and not dummy variables. We address this comment by using an alternative approach proposed in literature to handle qualitative data set.

We incorporate the discussion in the main text as follows (page 8 Line 5):

“Whereas, the MCA index was generated by assigning ranks to each of the ten questions according to their relevance and attaches weights to each question to create a weighted sum (see Rencher and Christensen , 2012 or Everitt and Dunn , 2001). In the current literature, principal components or factor analysis (PCA) is most widely used for the construction of such indices. However, PCA is designed to handle quantitative data since it assumes a normal distribution of indicator variables. In contrast, multiple correspondence analysis (MCA) makes fewer assumptions about the underlying distributions of indicator variables and is more suited for qualitative data”.

2. Secondly, the authors should be upfront in explaining the anthropometric factors and grouping which drive the differences such as dietary intake, and ethnicity/race, and should further explain the caveats while relying on self-reported data such as highest and weight. Respondents can create systematic bias while providing such data. This can create an impact on the estimates, therefore, you should write about it, to make your reader aware of the possible shortcomings.

Author’s Response: Again an important point raised by the reviewer, we discuss the caveats of relying on the anthropometric measures for our analysis (Footnote 5).

“There are several caveats to interpreting the results associated with height for age z scores and weight for age z scores of children. Such as, the nutritional intake habits of different societies as well as genetic and ethnic diversity that on average can create differences amongst the children in those communities compared to other communities. In our analysis, we carefully introduce an exogenous variation in both the empowerment of mothers and the presence of grandmothers to interpret what could have happened if they were compared to their respective counterfactuals.”

3. Thirdly, in the data section, I think the authors should make the readers aware of the methodology of data collection concerning MICS and PDHS. Even if both are population representatives of Punjab, still there is a possibility that the variation in the outcome variable is solely due to the assumption taken over the data collection methodology. Although, the idea is not to make a comparison because both samples are different in a way as one estimates the effect of women empowerment on child health outcomes and the other is for co-residing grandparents affecting child health outcomes but still there should be a section which explicitly discusses the differences in data collection methodologies in both datasets.

Author’s Response: We have added more information regarding the methodologies both surveys follow and how similar yet different they are from each other (Line 18 Page 7).

“Overall, the survey sampling strategies and methodologies used for the PDHS and MICS are similar in many ways, but they differ in terms of the complexity of their sampling methods. The focus of PDHS is to collect data on the reproductive females and their health characteristics. It uses a simpler two-stage cluster sampling approach with households used as primary sampling units (PSU). While the MICS uses a more complex multistage cluster sampling approach and the focus of the data collection is to gather information on the wellbeing of the households. MICS sometimes is argued to be more comprehensive and complex that accurately collects information to give precise estimates.”

4. Lastly, one minor observation is, the study used three different samples, (i) measuring the impact of having at least one grandparent in the household, (i) measuring the impact of having only a grandfather in the household, and (iii) measuring the impact of having only the grandmother in the household on the health outcomes of children in a household but the title of the paper only reflects the paternal grandmothers?

Author’s Response: A very valid point raised, unfortunately, we do not have data to back our proposition that in a Pakistani society most of the couples reside in their husband’s houses and therefore much larger probability of co-resident paternal grandmother than maternal grandmother. Anyhow, we have used the word grandmother so that we do not make any distinction in terms of whether they share their relation with the kid due to the father or the mother.

Reviewer #2: 1. In the abstract it is a good idea to include some numbers from the result section to indicate the effect size of mother empowerment and presence of grandmothers with respective confidence intervals (p-values).

Author’s Response: According to the recommendation of the reviewer, we have added more content in the abstract discussing the average results in magnitude and significance. 

“In this paper, we show that i) empowered mothers and ii) coresident grandmothers each benefit children’s nutritional health measured by height-for-age z-scores (HAZ) and weight-for-age z-scores (WAZ) of the age group 5 years and less. First, using a cross-sectional Pakistan Demographic and Health Survey (PDHS) for the survey year 2017-18, we estimate the impact of empowered mothers on child health outcomes using an instrumental variable approach to correct for endogeneity. Empowerment is measured by two indices constructed separately: as a sum and alternately using multiple correspondence analysis (MCA), using the questions that gauge both intrinsic and extrinsic dimensions of female agency. Second, we use a fuzzy regression discontinuity design (FRDD) to measure the causal impact of coresident grandmothers on the health outcomes of the children using multiple rounds of the Multiple Indicator Cluster Survey (MICS, survey years 2008, 2011, 2014 and 2018). The difference between the actual ages of the grandmother from the Potential Retirement Eligibility Criteria (PREC) has been used to exogenously gauge the availability of the grandmother’s presence to the household. The results show that on average, the weight for age z scores (WFA) for children under five increases by 0.28 SD with one-index point increase in mother’s empowerment. Similarly, on average, a significant increase in WFA by 0.0984 SD is associated with the presence of grandmothers (alone) in a household. Finally, we explore heterogeneity in the average effect stated above based upon the gender, wealth and geographic location of the household. The benefits of mothers’ empowerment are largely driven by improvements in girls’ nutrition as well as children living in rural areas while the presence of grandmothers primarily improve the nutrition of boys, children in rural areas and belonging to poor families.”

2. In the introduction section first paragraph apart from the global statistics of stunting and wasting, it will be more valuable if such numbers are also given for Pakistan and specifically Punjab to get a good idea about the current situation.

Author’s Response: We thank reviewers for identifying the importance of putting some figures that could show us the health status of the children in Pakistan. We address the comment as follows (Page 3 Line No. 7):

According to WHO and United Nations International Children’s Emergency Fund (UNICEF), the nutritional health of children in Pakistan is of a big concern. When compared to its counterparts Bangladesh and India in Figure 1, we see that the data for Pakistan show that although we see a declining trend in the prevalence of children being underweight and stunting for Pakistan. However, child stunting in Pakistan is worse than that in India and Bangladesh.

Figure 1: Prevalence of stunting, height for age and prevalence of underweight, weight for age (modeled estimate, % of children under 5).

Source: The World Bank Database for the years 1991-2020

3. “The mother’s empowerment index constructed in two different ways: first, we use the sum of positive responses given, and second, we use a principal component index (PCA) for ten survey questions measuring the behavioral and attitudinal dimensions of daily choices mothers make within the household”. Why have you used two different measures for mother’s empowerment given same questions are used for both. What is the justification for using additive approach, will it be not biased as some factors are more important than others to measure empowerment. Secondly, after running the principal component analysis (PCA), which criteria was followed in context of the eigen values? What percentage of variation was covered by the principal component?

Author’s response: In literature, researchers have used Principal Component Analysis as one of the potential ways to reduce the dimensionality of the research question and create indices based upon multiple questions; hence, it was used in the analysis earlier. However, the reviewers correctly identified that PCA is used for continuous variables and not dummy variables. We address this comment by using an alternative approach proposed in literature to handle qualitative data set (page 8 Line 5):

“Whereas, the MCA index was generated by assigning ranks to each of the ten questions according to their relevance and attaches weights to each question to create a weighted sum (see Rencher and Christensen , 2012 or Everitt and Dunn , 2001). In the current literature, principal components or factor analysis (PCA) is most widely used for the construction of such indices. However, PCA is designed to handle quantitative data since it assumes a normal distribution of indicator variables. In contrast, multiple correspondence analysis (MCA) makes fewer assumptions about the underlying distributions of indicator variables and is more suited for qualitative data”.

4. PDHS survey was conducted in 2017-18, please write the years in this format rather than writing 2018 alone.

Author’s Response: Corrected.

5. “We use data from the household surveys conducted in Punjab in Pakistan (MICS has also collected information about two other provinces in Pakistan, Sindh and Baluchistan. In the case of Baluchistan, the quality of data is poor, especially with regards to the nutrition indicators, and therefore not much useful information can be extracted from those surveys.)”. This explains why Baluchistan was not selected for analysis but no reason is given for Sindh? However, it would be interesting if difference between Punjab and Sindh is observed for the role of grandparents given MICS collect reliable data for Sindh.

Author’s Response: We do not have access to this dataset. It would require a fair share of time to clean the data and re do the analysis that we have done in this paper. However, it will be a good analysis for future studies.

6. “Where ℎ is the dependent variable that comprises the nutritional health for children aged 5 years and younger”. However, there is no explanation given what the subscript g,h and i means for the dependent and independent variables. Please include the level which it is representing.

Author’s response: We have carefully incorporated the description of each of these subscripts under each respective specification. 

Page 8 “where Y_ghi is the dependent variable that comprises the nutritional health of child i in h is the household level living in district g for children aged 5 years and younger.”

Page 12 “One can argue that having more children in a household might impose stricter resource constraints, which in turn can affect the health outcomes of child i in the age group under 5 years in a household h in district g.”

Page 13 “The equation above uses fitted probabilities from equation (2) to instrument for female empowerment to correct for the endogenous “empowerment of mother” variable that affects the anthropometric measures of child i in a household h located in district g.”

7. What is the reason behind using square term for the child age, mother’s age, household head age and wealth index in addition to their level form? There is no interpretation given of the coefficient of these square terms and what does its results signify in this manuscript?

Author’s Response: An important point raise by the reviewer. To address the non-linear relation between these variables and the anthropometric measures. We report the coefficients of these measures in the appendices. We add the relevance of squared terms in the main text (Page 10 Line 21) and interpretation of their confidents as well (Page 26 and page pg33).

Page 10

“We control for the age squared of the child and the mother in the regression to capture for any non-linearity in the relation between the age variables and its relation to the health outcomes of children.”

Page 26” We see that on average younger children tend to be significantly less healthy and as their age increases the impact on tend to have children the health outcomes improve. . Similarly, we see that the child’s health is significantly lower if they have a younger mother and that it increases significantly as the age of the mother increases after a threshold age.”

Page 33 “Younger mothers tend to have children with lower health outcomes as compared to mothers above a threshold age indicated by the mother’s age squared coefficient.”

8. “Therefore, to address this problem, we use the number of sons a woman has as an instrument for mother’s empowerment. We argue that the number of sons a woman has is correlated with the empowerment of a woman, especially in the context of South Asia, where sons are given special importance in a household but are orthogonal to the health outcomes of children aged 5 years and less in a household”. There is no backing of literature that number of sons is a good instrument for mother’s empowerment and how it is not correlated directly to health outcomes of children aged 5 years and less. Number of sons can also have a direct impact on the health outcome of children, parents might be more concerned about the health of boys and provide them better nutrients and health care as there exists a preference of boys in the South Asian countries. The theoretical framework is lacking for the choice of instrument in the manuscript. Please add the relevant literature to support the choice of this IV which should justify it.

Author’s Response: A very valid point raised by the reviewer. We have added the logical explanation and cited literature to explain why in some south Asian societies; the mother’s empowerment may increase as she gives birth to more sons (Page 12 Line No. 1).

“Alfano (2017) argues that women with lesser control over household income secure a stronger bargaining position by relying more on their male offspring. This point is further reinforced in literature by arguing that after certain age of father, mothers gain more power in terms of taking decision in a household as they become more loyal to the future decision makers of the households i.e., their sons (Gupta et al., 2003; Zimmermann, 2018).”

9. What kind of wealth index is used in this study? Is it the DHS wealth index already developed in the dataset or some other type of wealth index is used? Please specify it in the manuscript. Also, for the MICS data set, which wealth index is used?

Author’s Response: We have added the explanation of generating wealth indices in both of the surveys (Footnote 6).

In the case of PDHS, the wealth index is created using Principal Component Analysis (PCA) that use various indicators like household characteristics, durable goods and assets to determine the pattern of wealth amongst the households. They later divide the households amongst wealth quintiles. The lowest wealth quintile represents the 20% of the population that is part of the most constrained households (i.e. Poorest). We use the bottom two quintiles to identify the population that is severely constrained and most severely constrained in resources and later argue that these households may have been affected differently due to the empowerment of mothers.

10. “Dummy=1 if the mother ever used contraceptives in table 1 b”. This variable is showing the minimum and maximum value of 0 and 8. However, given this is a binary variable the maximum should be equal to 1. Can you please clarify on this?

Author’s response: We have corrected for this error in the table. We have replaced the missing values coded as 8 to a missing value. It has changed the mean value and standard deviation in the descriptive statistics in fourth decimal place.

Dummy=1 if the mother ever used contraceptives 182,610 0.1594747 0.3676328 0 1

11. In table 1b, the minimum value of mother’s age is 15 and the maximum value for number of years married is 43. These minimum and maximum values are not looking realistic, please into it and provide explanation for it.

Author’s response: PDHS and MICS (for the under 5 roaster) collects information on women who are in their fertile window and have children of age 5 or less. Therefore, they donot gather information of any mother who does not qualify into this category.

12. Apart from empowered mother and presence of grandparents, results of other independent variables are not reported and interpreted in the manuscript. Some of these independent variables are of interest which should be included in the manuscript for completeness of results. For example, given the empowerment level of mother, what is the effect of household head’s education level on the children health outcome?

Author’s response: A very valid point raised by the reviewers. We have added the table with all the covariates controlled in the specification in Appendix A: Table A1 and A2 and reported the main impacts in other control variables sections 3.1.4 (Page 26) and 3.2.3 (Page 33).

Section 3.1.4 Other Controls

“We see few other important affects as show by the control variables on the health outcome on the children (Appendix A, Table A1). We see that on average younger children tend to be significantly less healthy and as their age increases the impact on tend to have children the health outcomes improve. Similarly, we see that the child’s health is significantly lower if they have a younger mother and that it increases significantly as the age of the mother increases after a threshold age. Similarly, mother’s education has a significant positive impact on the health outcomes of the children. We see on average boys health outcomes are significantly lower if the child was a girl. Working women, on average tend to have children with significantly lower health outcomes. Likewise, we see that children in lower wealth quintiles compared to top wealth quintile have children with significantly lower health outcomes.”

Section 3.2.3 Other Controls 

“We report the results of the entire specification measuring the impact of all the controlled variables in addition to the presence of grandmothers in a household on the health outcomes of the children in Appendix A, table A2. We see similar impacts as that we have reported in section 3.1.4 of child’s characteristics, mother’s characteristics and household characteristics on the child’s health outcomes. The results show that on average the girls are healthier than boys are, the positive squared term of age shows that as the age of the child increases they become healthier. Children in urban areas are healthier than the children located in rural areas. Child’s health improves as the wealth of the household improves. Younger mothers tend to have children with lower health outcomes as compared to mothers above a threshold age indicated by the mother’s age squared coefficient.

13. Why separate regressions are estimated for region (urban/rural), gender (boys/girls) and wealth status (poor/wealthy)? Why interaction terms were not added with empowered mother and presence of grandparents to analyze the effect of region, gender and wealth with the impact variable (empowered mother and presence of grandparents)? Split regression does not account for heterogenous effect accurately as the significance depends on other covariates as well.

Author’s Response: We thank the reviewers for this comment. We have revised all the regressions and have used the interaction analysis instead of using subsample regressions. To the best of our understanding, the results estimated from sub-sample regressions are similar to the interaction analysis as long as the covariates in both of the regressions are same. However, we see a slight change in magnitude and improvement in standard errors while using interaction analysis. Therefore, our results have improved by using the specifications recommended by the reviewers.

Table 4: Measuring the Impact of the Mother’s Empowerment on the Child’s Health Outcomes by Rural/Urban Divide

 Empowerment measured using Additive Index Empowerment measured using MCA

 OLS

(1) OLS with Controls

(2) IV

(3) IV with Controls

(4) OLS

(1) OLS with Controls

(2) IV

(3) IV with Controls

(4)

Dependent variable Weight for age Z-Scores

Mother's empowerment Index 0.017** -0.019** 0.188 0.622** 0.047 -0.071** 4.418 1.333**

 (0.008) (0.009) (0.140) (0.261) (0.033) (0.031) (140.546) (-0.091)

empowerment *urban 0.028*** 0.018 -0.003 -0.374 0.171*** 0.074* -6.128 -1.347

 (0.007) (0.013) (0.029) (0.392) (0.048) (0.045) (225.460) (1.347)

Observations 4,606 4,604 4,606 4,604 4,606 4,604 4,606 4,604

R-squared 0.020 0.146 0.019 0.146 

1st F-test 18.601 24.161 20.089 24.289

 Dependent variable Height for Age Z-Scores

Mother's empowerment Index 0.013* -0.009 0.203 0.174 0.034 -0.029 -7.714 0.720

 (0.007) (0.008) (0.150) (0.202) (0.028) (0.029) (29.239) (0.741)

empowerment* urban 0.005 0.008 -0.035 0.065 0.061 0.025 11.647 0.122

 (0.006) (0.012) (0.028) (0.287) (0.042) (0.042) (43.896) (0.955)

Observations 5,158 5,156 5,158 5,156 5,158 5,156 5,158 5,156

R-squared 0.003 0.041 0.004 0.041 

1st F-test 4.264 7.053 5.273 7.056

Note: The two dependent variables are height-for-age z scores and weight-for-age z scores for the children of age group 5 years and less. The main independent variable, mother’s empowerment is measured by indices constructed in two different ways: first, the additive index, and second, the index created by the multiple correspondence analysis (MCA). Controls include the child’s characteristics: gender, age, and age squared. Household characteristics: urban, gender of the household head, total number of households, wealth score, household head education level, wealth score square. Mother’s characteristics: mother’s education level, mother’s age, mother’s age squared, age of the first born, number of years married. The geographical controls comprise district and province fixed effects. Standard errors are clustered at the household level. *** p<0.01, ** p<0.05, * p<0.1

Table 5: Measuring the Impact of Mother’s Empowerment on the Child’s Health Outcomes by gender

 Empowerment measured using Additive Index Empowerment measured using MCA

 OLS

(1) OLS with Controls

(2) IV

(3) IV with Controls

(4) OLS

(1) OLS with Controls

(2) IV

(3) IV with Controls

(4)

Dependent Variable Weight for Age Z-Scores

Empowerment Additive Index 0.031*** -0.008 0.187** 0.401 0.132*** -0.031 0.602 1.576

 (0.007) (0.008) (0.090) (0.294) (0.028) (0.029) (1.394) (1.210)

empowerment*boy 0.004 -0.007 0.001 0.110 -0.025 -0.023 0.056 0.386

 (0.004) (0.008) (0.008) (0.190) (0.031) (0.029) (3.138) (0.682)

Observations 4,606 4,604 4,606 4,604 4,606 4,604 4,606 4,604

R-squared 0.012 0.145 0.013 0.145 

1st F-test 11.988 24.183 12.938 24.168

Dependent Variable Height for age Z-Scores

Empowerment Additive Index 0.020*** -0.007 0.119 0.260* 0.060** -0.027 1.002 0.961*

 (0.006) (0.008) (0.085) (0.152) (0.026) (0.027) (0.861) (-0.562)

empowerment*boy -0.007 0.002 -0.006 -0.169 -0.001 0.014 -1.443 -0.569

 (0.004) (0.008) (0.007) (0.130) (0.029) (0.029) (1.870) (0.441)

Observations 5,158 5,156 5,158 5,156 5,158 5,156 5,158 5,156

R-squared 0.003 0.041 0.003 0.041 

1st F-test 5.042 7.068 4.213 7.098

Note: The two dependent variables are height-for-age z scores and weight-for-age z scores for the children of age group 5 years and less. The main independent variable, mother’s empowerment is measured by indices constructed in two different ways: first, the additive index, and second, the index created by the multiple correspondence analysis (MCA). Controls include the child’s characteristics: gender, age, and age squared. Household characteristics: urban, gender of the household head, total number of households, wealth score, household head education level, wealth score square. Mother’s characteristics: mother’s education level, mother’s age, mother’s age squared, age of the first born, number of years married. The geographical controls comprise district and province fixed effects. Standard errors are clustered at the household level *** p<0.01, ** p<0.05, * p<0.1.

Table 6: Measuring the Impact of Mother’s Empowerment on the Child’s Health Outcomes by wealth

Dependent Variable Empowerment measured using Additive Index Empowerment measured using MCA

 OLS

(1) OLS with Controls

(2) IV

(3) IV with Controls

(4) OLS

(1) OLS with Controls

(2) IV

(3) IV with Controls

(4)

 Weight for Age Z-Scores

Mother's empowerment Index -0.013 -0.017* -0.004 0.543 0.048 -0.057 7.801 1.128

 (0.009) (0.010) (0.143) (0.391) (0.036) (0.035) (10.897) (0.476)

Empowerment*rich 0.075*** 0.012 0.084*** -0.198 0.158*** 0.034 -10.375 -0.857

 (0.007) (0.013) (0.028) (0.568) (0.049) (0.046) (17.920) (1.959)

Observations 4,606 4,604 4,606 4,604 4,606 4,604 4,606 4,604

R-squared 0.068 0.145 0.065 0.018 0.145 

1st F-test 77.950 23.914 21.706 23.918

Dependent Variable Height for age Z-Scores

Mother's empowerment Index -0.010 -0.007 0.041 0.385 0.025 -0.024 25.047 1.443

 (0.007) (0.009) (0.183) (0.466) (0.029) (0.030) (402.348) (1.588)

Empowerment*rich 0.042*** 0.004 0.042 -0.274 0.076* 0.010 -38.201 -1.029

 (0.006) (0.012) (0.039) (0.751) (0.043) (0.042) (634.749) (2.630)

Observations 5,158 5,156 5,158 5,156 5,158 5,156 5,158 5,156

R-squared 0.021 0.041 0.004 0.041 

1st F-test 27.090 7.045 5.734 7.043

Note: The two dependent variables are height-for-age z scores and weight-for-age z scores for the children of age group 5 years and less. The main independent variable, mother’s empowerment is measured by indices constructed in two different ways: first, the additive index, and second, the index created by the multiple correspondence analysis (MCA). Controls include the child’s characteristics: gender, age, and age squared. Household characteristics: urban, gender of the household head, total number of households, wealth score, household head education level, wealth score square. Mother’s characteristics: mother’s education level, mother’s age, mother’s age squared, age of the first born, number of years married. The geographical controls comprise district and province fixed effects. Standard errors are clustered at the household level. *** p<0.01, ** p<0.05, * p<0.1

Table 9: Measuring the impact of the presence of grandmothers in a household on children’s health outcomes by rural/urban divide

 OLS

(1) OLS with Controls

(2) IV

(3) IV with Controls

(4)

Dependent Variable Weight for Age Z-Scores

Dummy=1 if the households have only Grandmother 0.003 -0.004 0.054 0.160***

 (0.022) (0.028) (0.042) (0.057)

Presence of Grandmother only * Urban 0.210*** -0.054 0.241*** -0.178*

 (0.038) (0.047) (0.074) (0.091)

Observations 187,918 99,218 187,918 99,218

R-squared 0.000 0.114 

1st F-test 24.566 238.396

Dependent Variable Height for Age Z-Scores

Dummy=1 if the households have only Grandmother 0.004 -0.023 0.019 0.079

 (0.027) (0.035) (0.051) (0.068)

Presence of Grandmother only * Urban 0.263*** 0.001 0.399*** 0.010

 (0.046) (0.054) (0.091) (0.108)

Observations 184,124 98,229 184,124 98,229

R-squared 0.000 0.125 

1st F-test 25.771 292.298

Note: The dependent variables are weight-for-age (WFA) and height-for-age (HFA). The independent variable comprises the main dummy variable, which takes a value of 1 if the grandmother is present in a household. Specifications (1) and (2) report OLS results, whereas specifications (3) and (3) report the IV results. The instrument used in the specification is Retirement Eligibility, which is grandmother’s age minus the potential retirement age (55 years for females). The controls include the child’s characteristics: gender, age, and age squared. Household characteristics: urban, gender of the household head, total number of households, wealth score, household head education level, wealth score square. Mother’s characteristics: mother’s education level, mother’s age, mother’s age squared, age of the first born, number of years married, district fixed effects, year fixed effects, Mothers age at the birth of child, dummy=1 if there are 2 children and above in a household, dummy=1 if there are 3 children and above in a household, dummy=1 if the second and above child is a girl, dummy=1 if the third and above child is a girl. Standard errors are clustered at the household level. *** p<0.01, ** p<0.05, * p<0.1

Table 10: Measuring the Impact of the presence of grandmothers in a household on children’s health outcomes by gender

 OLS

(1) OLS with Controls

(2) IV

(3) IV with Controls

(4)

Dependent Variable Weight for Age Z-Scores

Dummy=1 if the households have only Grandmother 0.070*** -0.030 0.137*** 0.169**

 (0.024) (0.032) (0.048) (0.067)

Presence of Grandmother only * Girl 0.019 0.013 -0.019 -0.120

 (0.034) (0.043) (0.068) (0.088)

Observations 187,918 99,218 187,918 99,218

R-squared 0.000 0.114 

1st F-test 9.919 238.170

Dependent Variable Height for Age Z-Scores

Dummy=1 if the households have only Grandmother 0.071** -0.047 0.152** 0.099

 (0.030) (0.038) (0.060) (0.078)

Presence of Grandmother only * Girl 0.055 0.051 -0.023 -0.033

 (0.042) (0.050) (0.084) (0.106)

Observations 184,124 98,229 184,124 98,229

R-squared 0.000 0.125 

1st F-test 10.773 292.232

Note: The dependent variables are weight-for-age (WFA) and height-for-age (HFA). The independent variable comprises the main dummy variable, which takes a value of 1 if the grandmother is present in a household. Specifications (1) and (2) report OLS results, whereas specifications (3) and (3) report the IV results. The instrument used in the specification is Retirement Eligibility, which is grandmother’s age minus the potential retirement age (55 years for females). The controls include the child’s characteristics: gender, age, and age squared. Household characteristics: urban, gender of the household head, total number of households, wealth score, household head education level, wealth score square. Mother’s characteristics: mother’s education level, mother’s age, mother’s age squared, age of the first born, number of years married, district fixed effects, year fixed effects, Mothers age at the birth of child, dummy=1 if there are 2 children and above in a household, dummy=1 if there are 3 children and above in a household, dummy=1 if the second and above child is a girl, dummy=1 if the third and above child is a girl. Standard errors are clustered at the household level. *** p<0.01, ** p<0.05, * p<0.1

Table 11: Measuring the impact of the presence of grandmothers in a household on children’s health outcomes by wealth

 OLS

(1) OLS with Controls

(2) IV

(3) IV with Controls

(4)

Dependent Variable Weight for Age Z-Scores

Dummy=1 if the households have only Grandmother -0.275*** -0.004 -0.280*** 0.184**

 (0.030) (0.041) (0.056) (0.075)

Presence of Grandmother only * Rich 0.520*** -0.029 0.644*** -0.128

 (0.037) (0.049) (0.070) (0.093)

Observations 187,918 99,218 187,918 99,218

R-squared 0.001 0.114 0.001 0.114

1st F-test 107.736 238.174

Dependent Variable Height for Age Z-Scores

Dummy=1 if the households have only Grandmother -0.326*** -0.028 -0.386*** 0.163*

 (0.039) (0.053) (0.069) (0.092)

Presence of Grandmother only * Rich 0.624*** 0.009 0.832*** -0.122

 (0.047) (0.061) (0.087) (0.111)

Observations 184,124 98,229 184,124 98,229

R-squared 0.001 0.125 

1st F-test 100.272 292.368

Note: The dependent variables are weight-for-age (WFA) and height-for-age (HFA). The independent variable comprises the main dummy variable, which takes a value of 1 if the grandmother is present in a household. Specifications (1), (3), (5) and (7) report the OLS results, whereas specifications (2), (4), (6) and (8) report the IV results. The instrument used in the specification is Retirement Eligibility, which is grandmother’s age minus the potential retirement age (55 years for females). The controls include the child’s characteristics: gender, age, and age squared. Household characteristics: urban, gender of the household head, total number of households, wealth score, household head education level, wealth score square. Mother’s characteristics: mother’s education level, mother’s age, mother’s age squared, age of the first born, number of years married, district fixed effects, year fixed effects, Mothers age at the birth of child, dummy=1 if there are 2 children and above in a household, dummy=1 if there are 3 children and above in a household, dummy=1 if the second and above child is a girl, dummy=1 if the third and above child is a girl. Standard errors are clustered at the household level. *** p<0.01, ** p<0.05, * p<0.1

14. In section 3.1.3, the household are divided into wealthy and poor households. Can you please elaborate on why bottom two quintiles are considered as poor and upper two quintile are categorized as wealthy household? What about the third quintile which lies in between the upper and lower two quintiles? If the DHS wealth index is considered in this study, then it is a relative measure of wealth calculation according to the given sample. The households which are falling in the bottom two quintiles might not be poor is absolute terms as the DHS wealth index divides the whole sample into 5 equal quintiles. How would this effect the categorization of household as poor or wealthy?

Authors’ response: We have added the explanation of how wealth quintiles are generated in PDHS and MICS. Also, we explain that the most constrained categories are the lowest two quintiles therefore, treating them as a different sub group (Footnote 7).

“The methodology for constructing wealth indices in the Multiple Cluster Survey (MICS) is similar to that used in PDHS. Though the variables used to construct the index using PCA in different rounds of survey might differ slightly, overall exercise ensures that the households are divided into quintiles based upon their wealth resources. The lowest two quintiles comprise of the 20% of the population referred as constrained and severely constrained, respectively.”

15. There is no discussion section present in the manuscript, which analyzes the results, provide its explanation and compare it with the literature. The results section is only interpretation the coefficients of the estimated models. Please add a discussion section in the manuscript where results are discussed and logical reasoning is given for the derived results while comparing it with international and national studies and recommending actions for better health outcomes of the children aged 5 years and below.

Authors’ response: We appreciate the reviewers for raising point since it has improved our understanding of the results.

First, we have incorporated the potential mechanisms explaining the results after every table.

Second, we have added more content in the conclusion section, where we briefly compare our results with the current literature.

Page 23 “The results show that empowered mothers are crucial for the betterment of the child’s health. Empowered mothers become capable of improving the health outcomes of their children due to the important decisions they may take differently as compared to the disempowered mothers. From providing nutritious food to utilizing better health services and observing improved hygiene and sanitation, empowered mothers can directly advocate for their child’s health and wellbeing. Due to which the empowered mothers are more likely to take precautions and can potentially raise their children with more care.”

Page 24 “We argue that under constrained resources the impact of empowered mothers become more pronounce. With limited access to health care services and poor education about health care and nutritional intake, empowered mothers may be able to suppress the impacts of these constraints on their child’s health and can find ways to combat them.”

Page 25 “These results not only advocate for better nutritional intake and improved decisions for hygiene and sanitation but also sheds light on the inter-generational impact in terms of empowered women giving birth to empowered girls that themselves will one day grow into empowered women. This will not only improve the overall health outcomes of the children but also encourage gender equality that may have far fetching impacts on the society.”

Page 29 “The results show that the presence of a grandmother plays a positive role in the lives of children in a household. These results can be attributed to the support, love and wealth of experiences that the grandmothers may provide to the children. The grandmothers can pass down traditional methods and different perspective for the childcare and could be considered as a wise council in a household. In addition, they can be a close substitute to the parents for short time periods that can help the parents to take a break from their hectic routine that may diffuse the tension in the household, which could further improve the child’s sense of belonging.”

Page 31 “The importance of grandmothers in rural areas may become more pronounced due to the limited resources available to the household. In such conditions, grandmothers can serve as a helping hand in the upbringing of the children and can provide financial and emotional support to them. Furthermore, they can serve as a strong role model that can pass traditional information and knowledge about childcare practices to improve the health outcomes of children.”

Page 31 “The results show that although there is no negative impact of the presence of grandmother on the health outcomes of girls, we see a significantly positive impact on the health outcomes of the boys implying that on average the boys receive more benefits due to the presence of grandmothers as compared to the girls. These results indicate certain amount of gender discrimination but at least not at the cost of the health of girls in a household.”

Page 32 “These results again reinforce the fact that the presence of grandmothers play a trivial role in households that are more vulnerable and constrained. Grandmother’s presence can facilitate the parents by offering childcare allowing them to work and concentrate outside of the house without having to take on the stress of taking care of their children in person.”

Page 34-35 “The literature is undivided when it comes to concluding that empowered mothers have a positive impact on the child health outcomes. What is not common is how to measure the empowerment of mother. Different ways have been proposed in literature to measure the empowerment of mother. For instance, using the size of mother’s social network to measure her empowerment, Moestue et al. (2007) showed in his study that was conducted on a small city in India that mothers with larger networks had access to wider range of resources that had a positive effect on length for age z-scores (LAZ) of their children. However, our analysis revolves around the definition of empowerment based upon the authority of the women over decision-making process with in the household. For selective autonomy that a woman may have in a household, Mashal et al. (2008) using limited set of questions to measure empowerment, reports positive impacts on stunting, wasting and child being underweight. He only incorporates the decision of the mother regarding the health care of children without permission to measure her empowerment. Similarly, Aslam & Kingdon (2012) reports positive impacts on the HFA z scores and WFA z scores if the mother is perceived to decide upon the number of children she wants to conceive. Shroff et al. (2009) reports that a mother has healthier children in terms of significantly higher HFA z scores, WFA z scores and LFA z scores, if she is actively engaged in taking the decisions of the child care, cooking and food supplies. Most of these either used limited set of questions to measure the empowerment of mother or used a simple linear regression framework to estimate the impact of empowerment on the child’s health outcomes. 

In this study, not only do we use an instrumental variable approach to correct for the endogenous empowerment of mother in a household but also take into account larger set of questions (intrinsic + extrinsic) to create an index of mother’s empowerment using PDHS 2017-18. This analysis not only show an improvement in the short-term measure of health outcomes (WFA z scores) for children in rural areas but also indicate that mother’s empowerment has ensured a long-term impact on the girl’s health outcome i.e. improvement in HFA z scores. These results shed light on the relevance of mother’s empowerment on the empowerment of future women and therefore acting as a potential driver of positive change in future generations.

As far as the relevance of grandmothers is concerned, we see that the literature is divided when it comes to measuring the impact of co-resident grandparents, specifically grandmothers on the health outcomes of the children. While we see substantial difference in the degree of involvement of grandparents in lives of their children, it has been accepted globally that grandparents influence the lives of their grandchildren. One strand of literature argues that over indulgence and division of constrained resources due to the presence of grandmothers may have a negative impact on the health outcomes of the children (Pearce et al., 2001; Watanabe et al., 2011). On the contrary, large body of literature reports positive impact of the presence of grandmothers on the health outcomes of the children due to two distinct reasons; first, providing informal child care and secondly, sharing wealth of knowledge and other resources such as inheritance and social networks (Pulgaron et al., 2013 & Aubel, 2012)). Numerous studies have reported obesity in the children living with grand mothers (Li et al., 2013; Polley et al., 2005; Tanskanen, 2013).

 Much of these studies have used qualitative analysis and thematic analysis to estimate the impact of the presence of the grandmothers on the child’s health outcomes. We use a more sophisticated technique ‘Fuzzy Regression Discontinuity Design” to cater to the problem of endogenous decision of grandmothers living with in a household. Our results reinforce on the relevance of grandmothers in a south Asian country especially, where the resources are limited and their presence can provide an informal childcare. Not only do we show that on average the short-term measure of health improves in the households with grandmothers but we also show that larger gains are associated with children categorized as being in ‘vulnerable group’ i.e., in rural areas and poorer households. However, we show that boys get significantly higher benefits from grandmothers compared to girls.

In addition to addressing the suggestions of the reviewers, we have thoroughly edited the paper to correct spelling and grammatical errors.

We look forward to hearing from you regarding our submission and please feel free to contact us if you have any additional suggestions or questions.

---

## [Decision Letter · Decision Letter 1]

7 May 2023

"Empowered Mothers and Co-Resident Grandmothers: Two Fundamental Roles of Women Impacting Child Health Outcomes in Punjab, Pakistan"

PONE-D-22-19975R1

Dear Dr. Arif,

We’re pleased to inform you that your manuscript has been judged scientifically suitable for publication and will be formally accepted for publication once it meets all outstanding technical requirements.

Kind regards,

Faisal Abbas, PhD

Academic Editor

PLOS ONE

Additional Editor Comments (optional):

Accept with minor revision no need to re-review.

Reviewers' comments:

Reviewer's Responses to Questions

**Comments to the Author**

1. If the authors have adequately addressed your comments raised in a previous round of review and you feel that this manuscript is now acceptable for publication, you may indicate that here to bypass the “Comments to the Author” section, enter your conflict of interest statement in the “Confidential to Editor” section, and submit your "Accept" recommendation.

Reviewer #1: All comments have been addressed

Reviewer #3: All comments have been addressed

2. Is the manuscript technically sound, and do the data support the conclusions?

Reviewer #1: Yes

Reviewer #3: Yes

3. Has the statistical analysis been performed appropriately and rigorously? 

Reviewer #1: (No Response)

Reviewer #3: Yes

4. Have the authors made all data underlying the findings in their manuscript fully available?

Reviewer #1: Yes

Reviewer #3: Yes

5. Is the manuscript presented in an intelligible fashion and written in standard English?

Reviewer #1: Yes

Reviewer #3: Yes

6. Review Comments to the Author

Reviewer #1: Thank you so much to authors for incorporating all the comments in the manuscript. I believe manuscript is now in much better shape to be published. Congratulation to the authors.

Reviewer #3: Comments:

1. The distribution of ℎ is not specified; it is important to check autocorrelation and independence of the estimated ℎ for the estimated regression model.

2. Did you check the normality of the variable of interest in this study through pp plots?

3. Use Adjusted R2 instead of R2 as a measure to explain the variation in the dependent variable explained by the independent variables.

4. Why the value of R2 is very low?

5. There are some grammatical mistakes in the write-up. Try to relook the paper with care.

7. PLOS authors have the option to publish the peer review history of their article (what does this mean?). If published, this will include your full peer review and any attached files.

Reviewer #1: **Yes: **SYED HASSAN RAZA

Reviewer #3: **Yes: **Dr. Tahir Abbas

---

## [Editor Report · Acceptance letter]

13 Jun 2023

PONE-D-22-19975R1 

Empowered Mothers and Co-Resident Grandmothers: Two Fundamental Roles of Women Impacting Child Health Outcomes in Punjab, Pakistan 

Dear Dr. Arif:

I'm pleased to inform you that your manuscript has been deemed suitable for publication in PLOS ONE. Congratulations! Your manuscript is now with our production department. 

Kind regards, 

on behalf of

Dr. Faisal Abbas 

Academic Editor

PLOS ONE